# Variable Clustering via Distributionally Robust Nodewise Regression

**Kaizheng Wang** [1 2]   **Xiao Xu** [1]   **Xun Yu Zhou** [1 2]

## Abstract

We study a multi-factor block model for variable clustering and connect it to regularized subspace clustering through a distributionally robust version of nodewise regression. To solve the latter problem, we derive a convex relaxation, provide a data-driven approach for selecting the size of the robust region, and develop an ADMM algorithm for efficient implementation. We validate our method in extensive numerical studies and demonstrate its superior performance.

## 1. Introduction

The rapid development of technologies has created an enormous amount of data in many fields. Such high-dimensional data often have many similar variables, in the sense that they convey a similar message and hence are replaceable with one another for certain tasks. It would then be useful to identify groups of similar variables and reduce the data complexity. This problem is called *variable clustering*.

Generally speaking, variable clustering is the problem of grouping similar components of a $d$-dimensional random vector $X = (X_1, \ldots, X_d)$. The resulting groups are referred to as *clusters*. In many applications, the problem of interest is to recover the clusters from a sample of $n$ independent copies, or observations, of $X$. This is essentially clustering the $d$ vectors, each having the $n$ observations. Variable clustering has been successfully applied to gene expression data (Jiang et al., 2004), protein profile data (Bernardes et al., 2015), financial data (Tang et al., 2022), among others.

A recent development in variable clustering is the $G$-block model proposed by (Bunea et al., 2020), which offers clearly defined population-level clusters. Under the $G$-block model,

the covariance matrix of $X$ has a block structure, and the blocks correspond to the clusters in a partition $G$, hence the name "$G$-block". In the $G$-block model, each cluster has one latent factor. Each variable is comprised of the factor in its cluster and an idiosyncratic component. Consequently, all variables in the same cluster are noisy realizations of the same latent factor. Since their observations lie near a single point in $\mathbb{R}^n$, it is natural to use centroid-based clustering approaches such as $k$-means (Bunea et al., 2020). A more flexible model is the multi-factor block model in (Ando & Bai, 2017), where each cluster may have several latent factors. Each variable is represented as a linear combination of its cluster-specific factors and an idiosyncratic component. Observations of variables in the same cluster lie near a low-dimensional subspace in $\mathbb{R}^n$ spanned by the same set of factors. All of the $d$ vectors in $\mathbb{R}^n$, each representing the observations of a variable, reside near a union of low-dimensional subspaces. Figure 1 shows such an example. The red circles represent variables in Cluster 1, which are approximate linear combinations of Factors 1 and 2. Hence, they are distributed near a plane. The blue triangles correspond to variables in Cluster 2, which are lined up along the direction of Factor 3. Consequently, the variable clustering

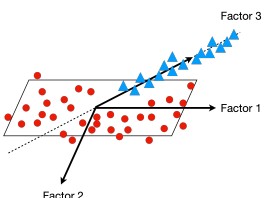

*Figure 1.* Subspace structure for variable clustering.

problem can be solved by identifying these subspaces and their corresponding variables. This task is usually referred to as *subspace clustering*, for which many techniques have been developed and applied to various real-world problems ranging from computer vision to machine learning (Vidal, 2011; Parsons et al., 2004).

A majority of common approaches for subspace clustering exploit the subspace structure by nodewise regression, where each variable is regressed against all other variables. The hope is that the regressions will favor the other variables in the same cluster over variables in different clusters. This way, the regression coefficients create an association matrix

---

[1]Department of Industrial Engineering and Operations Research, Columbia University, New York, NY 10027, USA [2]The Data Science Institute, Columbia University, New York, NY 10027, USA. Correspondence to: Kaizheng Wang <kaizheng.wang@columbia.edu>.

*Proceedings of the 43$^{rd}$ International Conference on Machine Learning*, Seoul, South Korea. PMLR 306, 2026. Copyright 2026 by the author(s).

that defines a weighted graph among variables. Then, the clusters can be recovered easily using, for example, spectral methods. When $d > n$, regularization is adopted in nodewise regression to make it well-posed. To that end, sparse subspace clustering (SSC), which adds $L_1$ regularization to the nodewise regression, is widely studied (Soltanolkotabi & Candés, 2012; Elhamifar & Vidal, 2013; Wang & Xu, 2016). However, a few drawbacks persist when using nodewise sparse regression for subspace clustering. First, tuning the parameter that controls the $L_1$ regularization depends on the unknown variance of the idiosyncratic components and is difficult. In addition, pursuing sparsity in the regression coefficients might be unnatural since the true association matrix can be dense as long as the subspaces are not orthogonal to each other or many variables in the same subspace have non-negligible correlations. To address these drawbacks, we propose a method that naturally incorporates regularization in the nodewise regression from the perspective of distributionally robust optimization (DRO). For a review on DRO, see (Rahimian & Mehrotra, 2022).

The main contributions of this article are the following.

- We connect a multi-factor block model for variable clustering to the subspace clustering problem. Based on that, we formulate a distributionally robust version of the commonly used nodewise regression method in subspace clustering. To our best knowledge, we are the first to apply DRO to nodewise regression and subsequently subspace clustering. This version of nodewise regression is motivated by the uncertainty in the data and leads to an *interpretable* regularization.

- We obtain a convenient convex relaxation of the distributionally robust nodewise regression, provide guidance on the choice of the size of the robust region, and propose an ADMM algorithm for efficient implementation. The algorithm significantly speeds up the calculation compared to off-the-shelf convex optimizers, which enables us to test it in high dimension. The superiority of our method compared with major peers is validated by extensive numerical experiments.

The rest of the paper is organized as follows. In Section 2, we provide a overview of variable clustering, subspace clustering, and nodewise regression. In Section 3, we introduce our DRO nodewise regression method and present theoretical results. Numerical experiment results are reported in Sections 4 to 6. The code is publicly available at https://github.com/xuxiao2695/dro-subspace-clustering.

## 2. Multi-Factor Block Model and Nodewise Regression

### 2.1. Problem setup

Given a $d$-dimensional random vector $X = (X_1, \ldots, X_d)$ and a sample of $n$ independent observations of $X$, we aim to find similar components of this random vector. First of all, consider the following single-factor block model, in which each random variable $X_i$ belongs to one of the $K$ clusters, indexed by $z(i) \in \{1, \ldots, K\}$. All random variables in the same cluster $k$ are associated with the same latent factor $F_k$. Formally,

$$X_i = F_{z(i)} + U_i,$$

where $\text{Cov}(F_{z(i)}, U_i) = 0$, $\text{Cov}(F) = \mathbf{\Sigma}_F$, and the idiosyncratic parts $U_i$ are uncorrelated, i.e., $\text{Cov}(U) = \mathbf{\Gamma}$ which is diagonal.

This single-factor block model naturally leads to the $G$-block model (Bunea et al., 2020). In the $G$-block model, the covariance matrix of $X$ has a block structure, with the blocks corresponding to groups of similar variables. Specifically, given a partition $G := \{G_1, \ldots, G_K\}$ of the variable indices $\{1, \ldots, d\}$ such that each $G_k$ is a set of $m_k$ indices, define the membership matrix $\mathbf{A} \in \mathbb{R}^{d \times K}$ associated with $G$: $a_{ik} = 1$ if $i \in G_k$ and $a_{ik} = 0$ otherwise. Suppose that $G$ is the true underlying cluster partition of the random variables $X_1, \ldots, X_d$. The model assumes that the covariance matrix $\mathbf{\Sigma}$ of the random vector $X \in \mathbb{R}^d$ follows a block decomposition, in which the blocks correspond to the groups in the partition $G$. This block structure means that variables in the same cluster have the same covariance with all other variables, and the covariance matrix $\mathbf{\Sigma}$ of $X$ can be decomposed as:

$$\mathbf{\Sigma} = \mathbf{A}\mathbf{\Sigma}_F\mathbf{A}^\top + \mathbf{\Gamma},$$

where $\mathbf{A}$ is associated with the partition $G$, $\mathbf{\Sigma}_F$ is a symmetric $K \times K$ matrix, and $\mathbf{\Gamma}$ is diagonal. When such a decomposition exists, we say that $X$ follows a $G$-block model.

The single-factor block model justifies methods where a single variable is used to represent the entire cluster. For example, the K-means algorithm essentially approximates $F_{z(i)}$ using the average of all $X_i$'s with the same $z(i)$, and (Tang et al., 2022) apply a variant of this model to cluster financial time series. The above model is arguably restrictive, as it assumes that each cluster is controlled by only one latent factor and all variables therein have the same loading. One may benefit from considering a more general model that allows the variables in the same cluster to be controlled by *a set of* factors. This motivates us to study the *multi-factor block model* (Ando & Bai, 2017), which is a natural extension of the single-factor block model.

Specifically, consider a $d$-dimensional random vector $X = (X_1, \ldots, X_d)$, and an underlying partition $G := \{G_1, \ldots, G_K\}$ of the indices $\{1, \ldots, d\}$. Denote by $m_k$

the size of cluster $G_k$. For each $k = 1, \ldots, K$, let $F_G^k$ be a $d_k$-dimensional random vector that represents the factors controlling the $k$-th cluster, and without loss of generality, assume that $\mathrm{Cov}(F_G^k) = \boldsymbol{I}$. We also assume that $m_k > d_k$, i.e., there are more variables than factors in each cluster. For each $i = 1, \ldots, d$, denote by $z(i) \in 1, \ldots, K$ the index of the cluster that $X_i$ belongs to.

**Definition 2.1** (Multi-factor block model)**.** Under the multi-factor block model, for each $i$, the random variable $X_i$ satisfies:

$$X_i = (F_G^{z(i)})^\top \beta_i + U_i, \tag{1}$$

where $\beta_i \in \mathbb{R}^{d_k}$ is the loadings of the $i$-th variable on the factors $F_G^{z(i)}$ and $U_i$ is a one-dimensional random variable that represents the idiosyncratic part satisfying $\mathrm{Cov}(U_i, U_j) = 0$ for $i \neq j$.

The above multi-factor block model is a special case of the general version in (Ando & Bai, 2017). The latter also includes observable global factors, which are not present in our problems of interest and thus omitted. From the multi-factor block model (1), we can see that the covariance matrix also displays a block structure and can be decomposed similarly to the $G$-block model.

*Fact* 1 (Multi-factor block model, matrix form). Let $F_G \in \mathbb{R}^D$ be the vector of all latent factors stacked together: $F_G := (F_G^{1\top}, \ldots, F_G^{K\top})^\top$, with $D := d_1 + \cdots + d_K$ being the total number of latent factors. We can write

$$\boldsymbol{\Sigma} = \boldsymbol{A}\boldsymbol{\Sigma}_F\boldsymbol{A}^\top + \boldsymbol{\Gamma},$$

where the $i$-th row of $\boldsymbol{A} \in \mathbb{R}^{d \times D}$ shows loadings of $X_i$ on all the $D$ factors: if $z(i) = k$, then $a_{i\cdot} = ( \overbrace{0, \ldots, 0}^{(d_1 + \cdots + d_{k-1})\ 0\text{'s}}, \beta_i^\top, \underbrace{0, \ldots, 0}_{(d_{k+1} + \cdots + d_K)\ 0\text{'s}} )$, $\boldsymbol{\Sigma}_F = \mathrm{Cov}(F_G)$, and $\boldsymbol{\Gamma} = \mathrm{Cov}(U)$.

A toy example illustrating the multi-factor block model and its induced near-block covariance structure is provided in Appendix D.4.

We remark that the variable clustering problem we consider is different from co-clustering (Dhillon, 2001; Dhillon et al., 2003). The latter assumes both the $d$ variables and the $n$ observations are clustered, aiming to simultaneously identify those two types of clusters. By contrast, our variable clustering problem does not require any cluster structure in the observations. They are often assumed to be i.i.d. from a continuous distribution (Bunea et al., 2020).

## 2.2. Subspace clustering and nodewise regression

Let $\boldsymbol{X} \in \mathbb{R}^{n \times d}$ be the data matrix of $n$ observations of $X$. Suppose that the (unobserved) realizations of the latent factors are $\boldsymbol{F}_G \in \mathbb{R}^{n \times D}$, then $\boldsymbol{X}$ can be decom-

posed as $\boldsymbol{X} = \boldsymbol{Y} + \boldsymbol{U}$, where $\boldsymbol{Y} = \boldsymbol{F}_G\boldsymbol{A}^\top$ is the group-specific component controlled by the factors, and $\boldsymbol{U}$ is the idiosyncratic components. Both $\boldsymbol{Y}$ and $\boldsymbol{U}$ are unobservable. One can see that the factor part of the $i$-th variable $y_i = \boldsymbol{F}_G^{z(i)}\beta_i$, which is an $n$-dimensional vector, lies in a $d_{z(i)}$-dimensional subspace, spanned by (unobserved) factor realizations $\boldsymbol{F}_G^{z(i)} \in \mathbb{R}^{n \times d_{z(i)}}$. For the subspaces to be meaningful, we assume that the number of observations is strictly larger than the maximum dimension of the subspaces, i.e., $n \geq d_k + 1$, $k = 1, \ldots, K$. Let $\mathcal{S}_k$ be the linear subspace of $\mathbb{R}^n$ spanned by the columns of $\boldsymbol{F}_G^k$, then each column of $\boldsymbol{Y}$ lies in the union of the $K$ subspaces: $\mathcal{S}_1 \cup \mathcal{S}_2 \cup \ldots \cup \mathcal{S}_K$. See Figure 1 for a visual illustration. Our goal is now to identify these $K$ subspaces from the data $\boldsymbol{X}$. As such, variable clustering under the multi-factor block model (1) is an instance of *subspace clustering* (Vidal, 2011; Parsons et al., 2004). Throughout the paper, we assume that $K$ is known. There are numerous methods for estimating $K$ in practice (Milligan & Cooper, 1985; Von Luxburg, 2007).

A common tool for subspace clustering is nodewise regression (Elhamifar & Vidal, 2013). Under the aforementioned subspace structure, each $y_i$ can be written as a linear combination of all other $y_j$'s that lie in the same subspace. To exploit the subspace structure of the group-specific components, it is then natural to regress each column $x_i$ of the data matrix $\boldsymbol{X}$ against all other $x_j$'s (hence the term "nodewise regression"). Specifically, for each $i = 1, \ldots, d$, we solve

$$\min_{b_i \in \mathbb{R}^d} \|x_i - \boldsymbol{X}b_i\|_2^2 \quad \text{s.t.} \quad b_{ii} = 0, \tag{2}$$

In matrix form, we can write equivalently

$$\min_{\boldsymbol{B} \in \mathbb{R}^{d \times d}} \|\boldsymbol{X} - \boldsymbol{X}\boldsymbol{B}\|_F^2 \quad \text{s.t.} \quad \mathrm{diag}(\boldsymbol{B}) = 0 \tag{3}$$

where $\|\cdot\|_F$ is the Frobenius norm of a matrix. The hope is that the resulting regression coefficients $\boldsymbol{B}$ will mainly connect vectors that are in the same cluster, i.e., $|b_{ij}| \approx 0$ if $z(i) \neq z(j)$. Then, the clusters can be easily recovered by performing, for example, spectral clustering on the symmetrized matrix $\boldsymbol{C} := \boldsymbol{B}_{abs}^\top + \boldsymbol{B}_{abs}$, where $(\boldsymbol{B}_{abs})_{ij} = |b_{ij}|$. This construction of the similarity matrix $\boldsymbol{C}$ from $\boldsymbol{B}$ is the standard practice in subspace clustering (Elhamifar & Vidal, 2013). The example in Appendix D.4 contrasts the covariance matrix and the similarity matrix extracted by population-level nodewise regression; the latter exhibits a much clearer block structure.

In summary, the subspace structure can be exploited through the following scheme:

1. Compute a similarity matrix $\boldsymbol{C}$ between vectors, ideally connecting only vectors in the same subspace with non-zero edges.

2. Construct clusters by applying spectral clustering techniques to $C$.

In the second stage, the spectral clustering algorithm in (Ng et al., 2001) is the most commonly used by the subspace clustering community, due to its simplicity and strong performance. One can also use other similarity-based clustering algorithms.

### 2.3. A review of nodewise regression

In this paper, we focus on step 1, specifically using nodewise regression to obtain the similarity matrix. In this section, we briefly review some existing variants of nodewise regression applied to subspace clustering and how nodewise regression connects to other areas of research.

In the existing literature, regularization of the regression coefficients is often added to the nodewise regression to overcome overfitting due to noise in the data and to deal with the issue of the regression (2) becoming ill-posed when $d > n$. One of the commonly used regularizers is the $L_0$ regularizer, which penalizes the number of non-zero regression coefficients. This $L_0$ semi-norm is usually relaxed to the $L_1$-norm as its tightest convex relaxation. The regression thus becomes the Lasso, which promotes sparse solutions and can be solved efficiently. Such subspace clustering methods using nodewise sparse regression for subspace clustering are called "sparse subspace clustering" (SSC, (Elhamifar & Vidal, 2013; Soltanolkotabi et al., 2014)). Others use nodewise regression with a nuclear-norm regularization, penalizing $\|\boldsymbol{B}\|_*$ in (3), thus encouraging it to be low-rank. This type of method is called "low-rank representation" (LRR) and is used in subspace clustering, segmentation, and feature extraction (Favaro et al., 2011; Liu & Yan, 2011; Liu et al., 2013; Chen & Yang, 2014).

Much of the current subspace clustering algorithms using nodewise regression can be improved. Take the Lasso-type SSC algorithm (Elhamifar & Vidal, 2013; Soltanolkotabi et al., 2014) as an example. SSC solves the following optimization problem. For every $j = 1, \ldots, d$,

$$\min_{b_j \in \mathbb{R}^d} \left\| x_j - \boldsymbol{X} b_j \right\|_2^2 + \lambda_j \left\| b_j \right\|_1 \quad \text{s.t.} \quad b_{jj} = 0, \quad (4)$$

for all $j = 1, \ldots, d$; or in matrix form,

$$\min_{\boldsymbol{B} \in \mathbb{R}^{d \times d}} \left\| \boldsymbol{X} - \boldsymbol{X} \boldsymbol{B} \right\|_F^2 + \left\| \boldsymbol{B} \boldsymbol{\Lambda} \right\|_1 \quad \text{s.t.} \quad \text{diag}(\boldsymbol{B}) = \boldsymbol{0},$$

$$(5)$$

where $\boldsymbol{\Lambda}$ is a $d \times d$ diagonal matrix whose diagonals are the parameters controlling the regularization $(\lambda_1, \ldots, \lambda_d)$. However, this approach has a few drawbacks. First of all, even under the true model, the coefficients are not necessarily sparse but usually a dense combination. Second, strong correlations among variables make $L_1$-based sparse recovery difficult. In addition, the appropriate $\lambda_i$ depends on

the variance of the idiosyncratic components, and heterogeneous and unknown variances of idiosyncratic components make tuning these parameters hard. As such, pursuing sparsity in the nodewise regression is unnatural and difficult to implement in practice. In comparison, we propose a method that naturally derives regularization in the nodewise regression by reformulating (3) in the context of distributionally robust optimization (DRO). This formulation results in a spectral-norm regularization. Importantly, the DRO analysis leads to an *endogenous* choice of the regularization parameter that is data driven, easy to compute and interpretable.

As a widely used technique in structural learning, the application of nodewise regression is not limited to subspace clustering. For instance, it is also closely related to the popular $k$-means clustering. The $k$-means algorithm for variable clustering (Bunea et al., 2020) amounts to the program

$$\min_{\mu_1, \ldots, \mu_K \in \mathbb{R}^n} \left\{ \sum_{i=1}^{d} \min_{z_i \in [K]} \|x_i - \mu_{z_i}\|_2^2 \right\}.$$

where $x_i \in \mathbb{R}^n$ is the observations of the $i$-th variable, $z_i$ is the index of the cluster that $x_i$ is assigned to, and $\mu_k$ is the mean of all variables in cluster $k$. According to the analysis of $k$-means in (Peng & Wei, 2007), this optimization problem can be reformulated as a nodewise regression problem with constraints:

$$\min_{\boldsymbol{B} \in \mathbb{R}^{d \times d}} \quad \left\| \boldsymbol{X} - \boldsymbol{X} \boldsymbol{B} \right\|_F^2,$$

$$\text{s.t.} \quad \boldsymbol{B} \in \left\{ \boldsymbol{Z} (\boldsymbol{Z}^\top \boldsymbol{Z})^{-1} \boldsymbol{Z}^\top : \right.$$

$$\left. \boldsymbol{Z} \in \{0, 1\}^{d \times K}, \ \boldsymbol{Z} 1_K = 1_d, \ \boldsymbol{Z}^\top 1_d > 0 \right\},$$

where $1_K$ and $1_d$ are all-one vectors with lengths $K$ and $d$, respectively; $\boldsymbol{Z}^\top 1_d > 0$ is an entrywise constraint.

Outside of clustering, the idea of nodewise regression has also been used in graphical model selection. For example, the authors of (Meinshausen & Bühlmann, 2006) use an $L_1$-norm-regularized nodewise regression to recover neighbors by estimating a sparse inverse covariance matrix. This technique is recently applied to Markowitz-type portfolio selection (Callot et al., 2019).

## 3. Distributionally Robust Nodewise Regression

In this section, we put the nodewise regression (3) in a probabilistic context. Consider the $d$-dimensional random vector $X$ whose coordinates have zero mean and unit variance. Denote by $\mathbb{P}^*$ the true probability measure underlying the distribution of $X$, and $\mathbb{E}_{\mathbb{P}^*}$ the expectation under $\mathbb{P}^*$. The classical least-square nodewise regression problem (3) is to solve:

$$\min_{\boldsymbol{B} \in \mathbb{R}^{d \times d}} \mathbb{E}_{\mathbb{P}^*} \left[ \left\| X - \boldsymbol{B}^\top X \right\|_2^2 \right] \quad \text{s.t.} \quad \text{diag}(\boldsymbol{B}) = 0 \quad (6)$$

Suppose that we have a data matrix $\boldsymbol{X} := (x_1, \ldots, x_n)^\top$, where $x_t \in \mathbb{R}^d$ is the $t$-th observation of the standardized random vector $X$. Denote by $\mathbb{P}_n$ the empirical distribution of the $n$ samples: $\mathbb{P}_n := \frac{1}{n} \sum_{t=1}^n \delta_{x_t}$. Given a cost function $c : \mathbb{R}^m \times \mathbb{R}^m \to [0, \infty)$ where $c(u, w) := \|w - u\|_2^2$ and two probability distributions $\mathbb{P}$ and $\mathbb{Q}$ supported on $\mathbb{R}^m$, we define the optimal transport cost or discrepancy between $\mathbb{P}$ and $\mathbb{Q}$, denoted by

$$
\begin{aligned}
\mathcal{D}_c(\mathbb{P}, \mathbb{Q}) = \inf \Big\{ &\mathbb{E}_\pi \big[ c(U, W) \big] : \pi \in \mathcal{P}(\mathbb{R}^m \times \mathbb{R}^m), \\
&\pi_U = \mathbb{P}, \pi_W = \mathbb{Q} \Big\} \\
= \inf \Big\{ &\mathbb{E}_\pi \big[ \|w - u\|_2^2 \big] : \pi \in \mathcal{P}(\mathbb{R}^m \times \mathbb{R}^m), \\
&\pi_U = \mathbb{P}, \pi_W = \mathbb{Q} \Big\}.
\end{aligned}
$$

The infimum is taken over all couplings between $\mathbb{P}$ and $\mathbb{Q}$. This discrepancy function is the squared Wasserstein distance of order two; it can be extended to any lower semicontinuous function $c$ such that $c(u, u) = 0$ for every $u \in \mathbb{R}^m$. As long as $c^{1/\rho}$ is a metric for some $\rho > 1$, $\mathcal{D}^{1/\rho}(\mathbb{P}, \mathbb{Q})$ is also a metric (Villani, 2009).

Recall that our original goal is to solve (6), which is the expected loss under the true distribution. The plug-in method, i.e., optimizing (6) under $\mathbb{P}_n$ generally yields unfavorable results that are poor out-of-sample, or under the true distribution $\mathbb{P}^*$. However, we cannot observe $\mathbb{P}^*$ but can only access the empirical distribution $\mathbb{P}_n$ inferred from the observations. The DRO approach is to postulate that $\mathbb{P}^*$ lies somewhere close to $\mathbb{P}_n$, e.g., within a region of radius $\delta$ around $\mathbb{P}_n$, leading to the following problem:

$$
\underset{\boldsymbol{B} \in \mathbb{R}^{d \times d}, \mathrm{diag}(\boldsymbol{B}) = 0}{\text{minimize}} \ \underset{\mathbb{P} : \mathcal{D}_c(\mathbb{P}, \mathbb{P}_n) \leq \delta}{\sup} \mathbb{E}_\mathbb{P} \Big[ \big\| X - \boldsymbol{B}^\top X \big\|_2^2 \Big]. \tag{7}
$$

By solving (7), we try to find coefficients $\boldsymbol{B}$ that optimize the *worst* regression error of (6) among all probability distributions within a region around $\mathbb{P}_n$. This region $\mathcal{U}_\delta(\mathbb{P}_n) := \{\mathbb{P} : \mathcal{D}_c(\mathbb{P}, \mathbb{P}_n) \leq \delta\}$ is called the uncertainty region with radius $\delta$ (Blanchet et al., 2019). If $\mathbb{P}^*$ indeed lies in this region, we are guaranteed that the loss under $\mathbb{P}^*$ will be no larger than what is achieved in (7).

At first glance, (7) appears very difficult to solve, as it involves the supremum over an (infinite-dimensional) space of probability measures. However, as we will show, this DRO problem can be relaxed as a finite-dimensional convex optimization problem. We also provide an ADMM algorithm that efficiently solves the latter. Finally, we provide a simple recipe for choosing the appropriate radius $\delta$ of the uncertainty region in Appendix B.

### 3.1. Transforming the DRO problem to convex optimization

The authors of (Blanchet et al., 2019) have presented the equivalence between distributionally robust linear regression with Wasserstein discrepancy of order $p$ and $L_q$ regularization, where $1 \leq p \leq \infty$ and $1/p + 1/q = 1$. Based on that, one might want to separate (7) into $d$ distributionally robust linear regressions and then solve their equivalent $L_2$-regularized formulations. However, this is not correct because the variables in those linear regressions are coupled. We will instead analyze the program (7) as a whole. The theorem below provides a convenient relaxation of the DRO problem (7) that is tight up to a factor of 2. The proof is deferred to the supplementary material.

**Theorem 3.1.** *With cost function $c(u, w) = \|w - u\|_2^2$, the following inequality holds for all $\boldsymbol{B} \in \mathbb{R}^{d \times d}$:*

$$
\frac{f(\boldsymbol{B})}{2} \leq \underset{\mathbb{P} : \mathcal{D}_c(\mathbb{P}, \mathbb{P}_n) \leq \delta}{\sup} \mathbb{E}_\mathbb{P} \Big[ \big\| X - \boldsymbol{B}^\top X \big\|_2^2 \Big] \leq f(\boldsymbol{B}),
$$

*where*

$$
f(\boldsymbol{B}) = \left( \frac{1}{\sqrt{n}} \|\boldsymbol{X} - \boldsymbol{X}\boldsymbol{B}\|_F + \sqrt{\delta} \|\boldsymbol{I} - \boldsymbol{B}\|_2 \right)^2,
$$

*and $\|\cdot\|_2$ represents the spectral norm of a matrix.*

Theorem 3.1 presents a relaxation of the DRO problem (7) that is equivalent to a convex program

$$
\underset{\substack{\boldsymbol{B} \in \mathbb{R}^{d \times d} \\ \mathrm{diag}(\boldsymbol{B}) = 0}}{\text{minimize}} \left\{ \frac{1}{\sqrt{n}} \|\boldsymbol{X} - \boldsymbol{X}\boldsymbol{B}\|_F + \sqrt{\delta} \|\boldsymbol{I} - \boldsymbol{B}\|_2 \right\}. \tag{8}
$$

The spectral norm penalty serves as a robustness regularizer to stabilize nodewise regression under uncertainty.

Our method naturally extends to distributionally robust *regularized* nodewise regression, such as the $L_1$-regularized version (5) for sparse subspace clustering. Specifically, the direct DRO formulation of (5) is

$$
\underset{\boldsymbol{B} \in \mathbb{R}^{d \times d}, \mathrm{diag}(\boldsymbol{B}) = 0}{\text{minimize}} \ \underset{\mathbb{P} : \mathcal{D}_c(\mathbb{P}, \mathbb{P}_n) \leq \delta}{\sup} \Big\{ \mathbb{E}_\mathbb{P} \Big[ \big\| X - \boldsymbol{B}^\top X \big\|_2^2 \Big] + \|\boldsymbol{B}\boldsymbol{\Lambda}\|_1 \Big\}.
$$

According to Theorem 3.1, a convex relaxation is

$$
\underset{\boldsymbol{B} \in \mathbb{R}^{d \times d}, \mathrm{diag}(\boldsymbol{B}) = 0}{\text{minimize}} \left\{ \left( \frac{1}{\sqrt{n}} \|\boldsymbol{X} - \boldsymbol{X}\boldsymbol{B}\|_F + \sqrt{\delta} \|\boldsymbol{I} - \boldsymbol{B}\|_2 \right)^2 + \|\boldsymbol{B}\boldsymbol{\Lambda}\|_1 \right\}.
$$

In fact, we may replace the $L_1$ penalty $\|\boldsymbol{B}\boldsymbol{\Lambda}\|_1$ with any convex function and still get a convex relaxation.

The regularization weight parameter $\delta$ is nothing but the diameter of the uncertainty region. To wit, the distributionally robust nodewise regression with squared loss (7) is approximately equivalent to a spectral-norm-regularized nodewise regression with square-root loss, where the strength of the regularization is controlled by the radius of the uncertainty region $\delta$. The spectral-norm (also called the operator norm) is widely used in the machine learning literature to describe the generalizability of a model by measuring its vulnerability against adversarial attacks (see, e.g., (Szegedy et al., 2014)). The same intuition applies to our problem. Let $l(x_t, \boldsymbol{B}) = \left\| x_t - \boldsymbol{B}^\top x_t \right\|_2^2$ be the regression loss for a given parameter $\boldsymbol{B}$ associated with an observation $x_t$. If $x_t$ is modified with some perturbation $\xi$, then we have

$$\frac{\left| l(x_t + \xi, \boldsymbol{B}) - l(x_t, \boldsymbol{B}) \right|}{\|\xi\|_2^2} = \frac{\left\| (\boldsymbol{I} - \boldsymbol{B})^\top \xi \right\|_2^2}{\|\xi\|_2^2} \leq \|\boldsymbol{I} - \boldsymbol{B}\|_2^2.$$

This means that the magnitude of relative changes in the loss compared with the magnitude of the perturbation can be bounded by the spectral norm of $\boldsymbol{I} - \boldsymbol{B}$. This interpretation is consistent with the intuition that DRO minimizes the worst-case loss inside a plausible region.

### 3.2. An ADMM algorithm

Program (8) is convex and can be solved by off-the-shelf optimizers. However, we find it prohibitively expensive in practice as soon as the dimension $d$ reaches the hundreds. We propose an efficient algorithm based on the alternating direction method of multipliers (ADMM) (Eckstein & Bertsekas, 1992; Boyd et al., 2010), which enjoys global convergence guarantees.

To begin with, we rewrite (8) as:

$$\min_{\boldsymbol{B}_1, \boldsymbol{B}_2 \in \mathbb{R}^{d \times d}} \left\{ \frac{1}{\sqrt{n}} \|\boldsymbol{X} - \boldsymbol{X}\boldsymbol{B}_1\|_F + \sqrt{\delta} \|\boldsymbol{B}_2\|_2 \right\},$$
$$\text{s.t.} \quad \boldsymbol{B}_1 + \boldsymbol{B}_2 = \boldsymbol{I}, \quad \text{diag}(\boldsymbol{B}_1) = 0.$$

We now describe the ADMM algorithm. Given an arbitrarily initialized $\boldsymbol{B}_2^0$ and $\boldsymbol{\Lambda}^0$, we repeat the following steps: At iteration $t$, update:

$$\boldsymbol{B}_1^{t+1} \leftarrow \operatorname*{argmin}_{\text{diag}(\boldsymbol{B})=0} \left\{ \frac{1}{\sqrt{n}} \|\boldsymbol{X} - \boldsymbol{X}\boldsymbol{B}\|_F \right.$$
$$\left. + \frac{\rho_t}{2} \left\| \boldsymbol{B} + \boldsymbol{B}_2^t - \boldsymbol{I} + \boldsymbol{\Lambda}^t \right\|_F^2 \right\} \quad (9)$$

$$\boldsymbol{B}_2^{t+1} \leftarrow \operatorname*{argmin}_{\boldsymbol{B}} \left\{ \sqrt{\delta} \|\boldsymbol{B}\|_2 \right.$$
$$\left. + \frac{\rho_t}{2} \left\| \boldsymbol{B}_1^{t+1} + \boldsymbol{B} - \boldsymbol{I} + \boldsymbol{\Lambda}^t \right\|_F^2 \right\} \quad (10)$$

$$\boldsymbol{\Lambda}^{t+1} \leftarrow \boldsymbol{\Lambda}^t + \boldsymbol{B}_1^{t+1} + \boldsymbol{B}_2^{t+1} - \boldsymbol{I},$$

where $\rho_t$ is a penalty parameter that can be fixed or adaptively adjusted over time; $\boldsymbol{B}_2^0$ and $\boldsymbol{\Lambda}^0$ are initialized as zeros. The above is repeated until the magnitudes of the updates are smaller than a predetermined threshold.

Each of the two sub-problems (9) and (10) are easily solved. Problem (9) is strongly convex due to the quadratic penalty and thus can be solved by first-order algorithms. Problem (10) has a partially closed-form solution based on the singular decomposition of $\boldsymbol{I} - \boldsymbol{B}_1^{t+1} - \boldsymbol{\Lambda}^t$, as stated in the following lemma.

**Lemma 3.2.** *Consider the optimization problem:*

$$\operatorname*{minimize}_{\boldsymbol{B} \in \mathbb{R}^{m \times n}} \quad \|\boldsymbol{B} - \boldsymbol{C}\|_F^2 + \lambda \|\boldsymbol{B}\|_2, \quad (11)$$

*where* $\boldsymbol{C} \in \mathbb{R}^{m \times n}$. *Let* $\boldsymbol{C} = \boldsymbol{U}\boldsymbol{\Sigma}\boldsymbol{V}^\top$ *be the singular value decomposition of* $\boldsymbol{C}$, *where* $\boldsymbol{U} \in \mathbb{R}^{m \times r}$, $\boldsymbol{V} \in \mathbb{R}^{n \times r}$, *and* $\boldsymbol{\Sigma}$ *is an* $r \times r$ *diagonal matrix whose diagonals are the singular values* $\sigma_1 \geq \sigma_2 \geq \cdots \geq \sigma_r \geq 0$ *of* $\boldsymbol{C}$. *Define* $\sigma_{r+1} = 0$. *Then the optimal solution* $\hat{\boldsymbol{B}}$ *can be expressed as* $\hat{\boldsymbol{B}} = \boldsymbol{U}\hat{\boldsymbol{S}}\boldsymbol{V}^\top$, *for some diagonal* $\hat{\boldsymbol{S}} \in \mathbb{R}^{r \times r}$ *whose diagonal* $s$ *satisfies for some* $k \in \{1, \ldots, r\}$, $s = \left( \overbrace{t, \cdots, t}^{k \text{ terms}}, \sigma_{k+1}, \sigma_{k+2}, \cdots, \sigma_r \right)$, *where*

$$t = \operatorname*{argmin}_{\sigma_{k+1} \leq u \leq \sigma_k} \left\{ \sum_{j=1}^{k} (\sigma_j - u)^2 + \lambda u \right\}.$$

We defer the proof of Lemma Lemma 3.2 to A.1 in the supplementary material. By virtue of this lemma, we can easily find the solution to (10) by computing the singular decomposition of $\boldsymbol{I} - \boldsymbol{B}_1^{t+1} - \boldsymbol{\Lambda}^t = \boldsymbol{U}\boldsymbol{\Sigma}\boldsymbol{V}^\top$, and then comparing the losses $\sum_{j=1}^{k} (\sigma_j - u)^2 + 2t\sqrt{\delta}/\rho$ among all $k \in \{1, \ldots, d\}$. The key idea here is to make use of nice properties of singular value decomposition (SVD) and spectral operations, which has also been employed by (Schönemann, 1966), (Lu et al., 2020), among others. The lemma shows that (10) effectively shrinks only the top singular values. One could consider approximate spectral routines, such as randomized SVD (Halko et al., 2011; Tropp et al., 2017), to avoid the computational cost of full SVD when $d$ is large.

## 4. Simulation Experiments

### 4.1. Subspace clustering methods for comparison

We first demonstrate our results through simulation. In the following experiments, we compare our DRO nodewise regression subspace clustering method (DRO) with the Lasso nodewise regression subspace clustering method (Lasso) as described in (4), Asset Clustering through Correlation (ACC) (Tang et al., 2022), the $k$-medoids algorithm ($k$-medoids) (Kaufman & Rousseeuw, 1990), multi-factor block model for clustering (MFC) (Ando & Bai, 2017), sparse subspace clustering (SSC) (Elhamifar & Vidal, 2013), elastic net subspace clustering (SSC-EnSC) (You et al.,

2016a), sparse subspace clustering by orthogonal matching pursuit (SSC-OMP) (You et al., 2016b), robust subspace segmentation by low-rank representation (LRR) (Liu et al., 2010), and co-clustering (Role et al., 2019).

- For DRO, the parameter $\delta$ is determined following the recipe described in Appendix B, where method (b) is used to calculate $\Upsilon_g$, and $M = 1000$ samples of $\boldsymbol{Z}$ are generated to determine the quantile, for which we set $1 - \alpha = 0.95$. We implement DRO through the ADMM algorithm in Section 3.2, and adopt the varying penalty parameter scheme in Section 3.4.1 of (Boyd et al., 2010), which adaptively adjusts $\rho_t$ during optimization based on the primal and dual residuals.

- For Lasso, we use a uniform parameter $\lambda$ for all regressions and determine the value for $\lambda$ using cross-validation by minimizing the validation error.

For the two methods above, the regression coefficients $\boldsymbol{B}$ are obtained then symmetrized by calculating $\boldsymbol{C} := \boldsymbol{B}_{abs}^\top + \boldsymbol{B}_{abs}$, where $(\boldsymbol{B}_{abs})_{ij} = |b_{ij}|$. Clusters are then calculated using the spectral clustering algorithm in (Ng et al., 2001) with $\boldsymbol{C}$ being the similarity matrix. Our construction of $\boldsymbol{C}$ from $\boldsymbol{B}$ and the clustering algorithm are consistent with the subspace clustering literature (Elhamifar & Vidal, 2013).

For ACC, we use a slightly modified version where the dissimilarity measure is

$$\text{CORD}(i, j) := \min \left( \max_{l \neq i, j} \left| \rho_{il} - \rho_{jl} \right|, \max_{l \neq i, j} \left| \rho_{il} + \rho_{jl} \right| \right)$$

in order to accommodate both positive and negative factor loadings. We also fix the number of desired clusters, instead of letting the algorithm decide. For $k$-medoids, we use the distance measure $1 - r^2$ with $r$ being the sample correlation between two variables. See Appendix D.1 for implementation details of the other algorithms.

## 4.2. Data generation

To generate synthetic data, we take a variant of (1) with an additional global factor:

$$X_i = \beta_H(i) F_H + F_{z(i)}^\top \beta_G(i) + U_i, \quad \text{for } i = 1, \ldots, d. \tag{12}$$

The global factor $F_H$ affects all variables and hence induces correlations among them explicitly. The new model can be rewritten in the form of (1) if we redefine $F_{z(i)}$ and $\beta_G(i)$ as $(F_H, F_{z(i)}^\top)^\top$ and $(\beta_H(i), \beta_G(i)^\top)^\top$, respectively. With given parameters $n$, $d$, $K$, $\beta_H(i)$, $d_k$, and $\text{Var}(U_i)$, the samples $\boldsymbol{X}$ are generated as follows. First, the sizes of the clusters, $\{m_k\}_{k=1}^K$, are determined following the multinomial distribution with equal probabilities $d/K$. For example, the first $m_1$ variables are marked as Cluster 1,

then the next $m_2$ Cluster 2. Then, a pool of $\min(n, d)$ candidate group-specific factors are generated as i.i.d. standard normal vectors. The direction of each factor is uniformly sampled from the unit sphere in $\mathbb{R}^n$. From this pool of candidate factors, $d_k$ group-specific factors are then randomly chosen for each cluster $k$. We note that two clusters may share one or more group-specific factors, as all clusters randomly pick factors from the same pool. Even if they do not, the two corresponding subspaces might not be orthogonal to each other since different group-specific factors can be correlated. Next, the factor loadings are determined. Loadings of variable $i$, represented by the $d_{z(i)}$-dimensional vector $\beta_G(i)$, are determined by sampling from the standard normal distribution. $\beta_G(i)$ is then normalized so that $\left\| \beta_G(i) \right\|_2^2 + \beta_H(i)^2 = 1$. Then, a hidden global factor $F_H$ is sampled from an $n$-dimensional standard normal distribution, and similarly $U_i$'s are drawn independently from a normal distribution with given variance $\text{Var}(U_i)$. In the end, the samples for each random variable are standardized to have zero mean and unit variance.

We create a total of $K = 25$ clusters among $d = 500$ variables. We let $\beta_H(i)^2$ and $\text{Var}(U_i)$ each be drawn independently and uniformly from $[0, 0.5]$. The higher $\beta_H(i)$ is, the less group-specific information there is in the data, and when $\beta_H(i) = 1$, there is no group-specific information. The number of factors controlling each cluster $k$ is randomly chosen from 1 to $m_k - 1$, where $m_k$ is the number of variables in cluster $k$. For each experiment, we generate $n = 250$ i.i.d. samples. Note that although our theoretical results for the DRO are stated when $n$ grows to infinity with $d$ fixed, we choose $n$ to be much smaller than $d$ to test the robustness of the DRO result. We run the experiment on 10 different random trials and examine the average Adjusted Mutual Information ("AMI", (Vinh et al., 2010)) between the obtained clusters and the ground truth. The AMI is a measure of similarity between two partitions; an AMI of 1 represents identical partitions, while uniformly random cluster assignments will have an AMI close to 0. The higher the AMI is for a partition compared with the ground truth, the more accurate the clustering results are.

## 4.3. Results

We visualize the true clustering structure, the sample correlation matrix, and the similarity matrices extracted by DRO and Lasso in Appendix D.5.

Table 1 shows the average AMI of each clustering method over the 10 random trials. Between the two subspace clustering methods, DRO achieves an average AMI of 0.92, followed by Lasso with an average AMI of 0.83. ACC and $k$-medoids underperform in this experiment, with average AMIs of 0.15 and 0.33, respectively. The underperformance of ACC and $k$-medoids is expected, because

*Table 1.* Average AMI of different clustering methods compared with ground truth, over 10 different random trials.

| DRO | Lasso | ACC | $k$-medoids | MFC |
|-----|-------|-----|-------------|-----|
| **0.92** | 0.83 | 0.15 | 0.33 | 0.43 |

| SSC | SSC-EnSC | SSC-OMP | LRR |
|-----|----------|---------|-----|
| 0.025 | 0.029 | 0.01 | 0.001 |

*Table 2.* AMI of different algorithms on the Extended Yale B dataset, with 20 additional random trials

| Metric | DRO | Lasso | $k$-medoids | MFC | ACC |
|--------|-----|-------|-------------|-----|-----|
| **Mean** | **0.580** | 0.403 | 0.084 | 0.172 | 0.006 |
| **Median** | **0.584** | 0.422 | 0.086 | 0.171 | 0.004 |

| Metric | SSC | SSC-EnSC | SSC-OMP | LRR | Co-Clust. |
|--------|-----|----------|---------|-----|-----------|
| **Mean** | 0.116 | 0.218 | 0.011 | -0.017 | 0.000 |
| **Median** | 0.118 | 0.220 | 0.012 | -0.018 | -0.001 |

they are not tailored to the subspace clustering problem: the model underlying ACC assumes variables from the same cluster are generated around the same single factor, and similarly, $k$-medoids only seeks points that are spatially close to each other. MFC stems from the same multi-factor model as ours, but its model fitting algorithm is not as accurate. The other algorithms have even worse performance. The average AMI of Co-Clustering is zero and hence omitted for space considerations.

To further understand how the clustering methods reacts to global factors of different magnitudes, we increase the noise level to $\mathrm{Var}(U_i) = 1$ and test $\beta_H^2(i) = 0.1, 0.2, \ldots, 0.9$ for all $i$, each value on 10 random trials. The average AMI of each method is shown in Figure 10 in Appendix D.5. The performances all decrease as the common factor becomes more dominant. DRO performs noticeably better than Lasso, while both outperform other methods again.

For completeness, we also examine the performance of the clustering algorithms with varying noise levels, homogenous group factor magnitudes, and no global factor. In those experiments, DRO also leads the cohort overall, and we include the detailed results and analysis in Appendix D.2. Finally, we report a representative wall clock runtime comparison in Table 11 (Appendix D.6), estimating $K = 25$ clusters among $d = 500$ variables over $n = 250$ observations. The ADMM algorithm reduces the runtime of the DRO clustering method by over 80% compared to off-the-shelf convex optimizers and is competitive with other clustering algorithms.

**Sensitivity analysis.** We conduct three ablation studies to examine the sensitivity of the DRO method. First, our ADMM uses an adaptive $\rho$-update scheme. To show its robustness, we vary the initial value $\rho = \rho_0$ across three orders of magnitude $(0.01, 0.1, 0.5, 1, 2, 5, 10)$. The results demonstrate robustness against initialization (average AMI ranges from 0.91 to 0.92). Second, we evaluate robustness to misspecification of the number of clusters $K$ by applying spectral clustering with $K \in \{10, 15, \ldots, 40\}$ to the DRO similarity matrix computed under the true $K = 25$. The method degrades gracefully: slight overestimation ($K = 27$) yields AMI $= 0.92$, comparable to the true $K$, while underestimation degrades performance more rapidly. Third, we test the confidence level $1 - \alpha$ used to calibrate $\delta$, sweeping $\alpha \in \{0.001, 0.01, 0.05, 0.1, 0.2\}$. The average AMI

remains between 0.91 and 0.93 across all values, confirming insensitivity to this choice. Full results with standard deviations are reported in Tables 8–10 in Appendix D.3.

## 5. Empirical Experiments on Face Clustering

In this section, we test the performance of the DRO with other clustering methods on the Extended Yale B dataset (Lee et al., 2005). The dataset consists of $192 \times 168$ pixel cropped face images of 38 individuals, with 64 frontal face images for each subject acquired under various lighting conditions. For each image, we first downsample it to the size of $24 \times 21$ pixels and reshape it into a 504-dimensional vector. According to (Basri & Jacobs, 2003), under the Lambertian assumption, images of a subject with a fixed pose and varying illumination lie close to a linear subspace of dimension 9. Therefore, one can assume that the vectors corresponding to all images in the dataset lie close to the union of 9-dimensional subspaces.

We adopt the same sampling methodology as in (Elhamifar & Vidal, 2013), dividing the 38 subjects into 4 groups, with the first three groups corresponding to subjects 1 to 10, 11 to 20, 21 to 30, and the fourth group corresponding to subjects 31 to 38. For each algorithm, we conduct three trials using the three sets of 10 subjects (1 to 10, 11 to 20, 21 to 30).

Table 2 reports the AMI of various clustering methods over 23 trials (three standard splits and 20 additional random trials). The DRO method achieves the best performance (mean 0.580, median 0.584). Results on the three standard splits only are reported in Appendix D.7.

## 6. Empirical Experiments on Financial Data

### 6.1. Overview

We now apply subspace clustering algorithms to financial time series data. Our task is to cluster the stocks in the S&P 500 universe using historical returns, and based on these clusters, construct stock portfolios. More specifically, we pick one representative stock from each cluster, and construct an optimized portfolio using these representative stocks. The underlying rationale is that by identifying stocks capable of representing others, one can create portfolios with a small number of stocks compared to the size of the

full universe, yet still achieve a sufficient level of diversification. See (Tang et al., 2022) for more detailed discussions on clustering and portfolio diversification.

## 6.2. Data preparation

We take the constituents of the S&P 500 as the universe. The data is obtained from Compustat through Wharton Research Data Services (WRDS), which consists of (1) the daily closing prices of the constituents; (2) the historical constituents data; and (3) the daily closing S&P 500 total return index with dividends reinvested, all between January 1996 and January 2020.

We apply clustering, stock selection, portfolio optimization, and backtesting for the period between February 2001 and January 2020. Partitions and portfolios are calculated on the first trading day of each February, starting with February 2001, in the then S&P 500 constituent stock universe. Specifically, at the end of the first trading day of each February, we choose the stocks in the S&P 500 Index according to the historical constituent data. Of all the current constituents, we discard stocks with less than 5 years of history and those with more than $5\%$ missing data in the past $n = 500$ days. If the same company has multiple classes of stocks in the S&P 500 Index (e.g. Alphabet Inc's GOOG and GOOGL), we only keep the class with the longest history. The numbers of eligible stocks that remained after the above filtering range between $468$ and $487$ over the backtesting period. For these eligible stocks, any missing prices are linearly interpolated using the previous and subsequent prices. Then, partitions are estimated based on the daily returns of the past $n = 500$ trading days. A smaller set of stocks are selected, and portfolios are constructed with optimized weights. These steps are described in detail in the following subsections.

## 6.3. Clustering and portfolio construction

We compare the following clustering approaches:

- **DRO-ACC:** First create $K_1$ clusters using the DRO subspace clustering algorithm, then split each cluster into $K_2$ sub-clusters using the ACC algorithm.

- **Lasso-ACC:** First create $K_1$ clusters using the Lasso subspace clustering algorithm, then split each cluster into $K_2$ sub-clusters using the ACC algorithm.

- **ACC, $k$-medoids, MFC, SSC, SSC-ENSC, SSC-OMP:** Create $K_1 \times K_2$ clusters using the corresponding algorithms.

- **LRR:** Create clusters using LRR without specifying the number of clusters.

See Appendix C.1 for a discussion. For each clustering method, once clusters have been constructed, we select the stock with the lowest volatility from each cluster and then form a portfolio on the resulting smaller set of stocks. Once a set of stocks is determined by the above procedure, we construct portfolios using the minimum variance allocation strategy to determine the weights of the stocks. As a benchmark portfolio, we take the S&P 500 Exchange Traded Fund (NYSE ticker: SPY), which is the largest ETF in the world and designed to track the S&P 500 Index. We refer to it as SPY. See Appendix C.2 in the supplementary material for more details of portfolio construction.

## 6.4. Results and analysis

We set $K_1 = K_2 = 6$. At each update in February, we find $K_1 \times K_2 = 36$ clusters, each of which contributes one stock in the portfolio. While this is a reasonable number of stocks to have in a portfolio, we also present results of $K_1 = K_2 = 3, 4, 5$ in Appendix C.4. In the results below, we update the portfolios once every year after the first trading day in February, when we re-do stock selection and re-compute allocation. Figure 2 in Appendix C.3 shows the cumulative performance of these portfolios in terms of the net value (starting at 1). The DRO-ACC portfolio outperforms the others significantly. Table 3 in Appendix C.4 reports the performance of the portfolios based on metrics commonly used in the wealth management industry.

To examine the compositions of the clusters, we compare them with sectors defined by the Global Industry Classification Standard (GICS)[1]. Figure 3 in Appendix C.3 shows the clustering results obtained by the DRO-ACC clustering method on Feb 1st, 2019, date of the last portfolio update in our experiment. The clusters are closely aligned with the GICS sectors. See Appendix C.3 for the detailed results.

## 7. Conclusion

In this paper, we propose a distributionally robust nodewise regression method and apply it to variable clustering. We derive a convenient convex relaxation of the problem. The uncertainty level in the distributionally robust regression can be chosen in a data-driven way. Compared with the popular sparse subspace clustering that uses nodewise Lasso, our method is tuning-free and has a naturally interpretable regularization. The only exogenous parameter of the algorithm is a confidence level $1 - \alpha$, which the algorithm is very insensitive to, as results are nearly identical with $\alpha = 0.01, 0.05, 0.1$. Simulation experiments show that our subspace clustering method outperforms many other methods in the literature. We also apply our method to face clustering (and to financial time series data for asset selection, provided in Appendix 6) and obtain promising and superior results.

---

[1]Available at https://www.msci.com/gics

## Acknowledgements

Kaizheng Wang acknowledges financial support through an NSF grant DMS-2210907 and a start-up grant at Columbia University. Xun Yu Zhou acknowledges financial support through a start-up grant and the Nie Center for Intelligent Asset Management at Columbia University.

## Impact Statement

This paper presents work whose goal is to advance the field of Machine Learning. There are many potential societal consequences of our work, none which we feel must be specifically highlighted here.

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

# A. Proof of Theorems

## A.1. Proof of Lemma 3.2

We know that the optimal solution $\hat{B}$ to (11) can be expressed as $\hat{B} = U\hat{S}V^\top$, for some $\hat{S} \in \mathbb{R}^{r \times r}$. Then we have:

$$\left\| \hat{B} - C \right\|_F^2 + \lambda \left\| \hat{B} \right\|_2 = \left\| \hat{S} - \Sigma \right\|_F^2 + \lambda \left\| \hat{S} \right\|_2.$$

To proceed, we need to use the following lemma.

**Lemma A.1** (Pinching Inequality (e.g., (Bani-Domi & Kittaneh, 2008))). *If a matrix $A$ has a block form:*

$$A = \begin{bmatrix} A_{11} & A_{12} & \cdots \\ A_{21} & A_{22} & \cdots \\ \vdots & \vdots & \ddots \end{bmatrix},$$

*then for any weakly unitary invariant norm $\|\cdot\|$,*

$$\|A\| \geq \left\| \begin{bmatrix} A_{11} & 0 & \cdots \\ 0 & A_{22} & \cdots \\ \vdots & \vdots & \ddots \end{bmatrix} \right\|.$$

Let $\tilde{S} := \mathrm{diag}\,(\hat{s}_{11}, \hat{s}_{22}, \ldots, \hat{s}_{rr})$. Because $\Sigma$ is diagonal and the Frobenius norm and the spectral norm are both weakly unitary invariant, by the pinching inequality, we have

$$\left\| \tilde{S} - \Sigma \right\|_F \leq \left\| \hat{S} - \Sigma \right\|_F, \quad \left\| \tilde{S} \right\|_2 \leq \left\| \hat{S} \right\|_2.$$

Because of the optimality of $\hat{S}$, we know that $\tilde{S} = \hat{S}$ and thus $\hat{S}$ is diagonal. Hence, (11) is equivalent to

$$\min_{S \in \mathbb{R}^{r \times r},\ S \text{ is diagonal}} \left\{ \|S - \Sigma\|_F^2 + \lambda \|S\|_2 \right\},$$

which is just

$$\min_{S \in \mathbb{R}^r} \left\{ \sum_{j=1}^r (s_j - \sigma_j)^2 + \lambda \max_j |s_j| \right\},$$

where $\sigma_1 \geq \sigma_2 \geq \cdots \geq \sigma_r \geq 0$ are singular values of $C$, and $s_1, \ldots, s_r$ are diagonal entries of $S$. This can be further transformed to

$$\min_{S, t} \left\{ \sum_{j=1}^r (s_j - \sigma_j)^2 + \lambda t \right\}$$
$$\text{s.t.} \quad 0 \leq s_1 \leq s_2 \leq \cdots, \leq s_r \leq t,$$

which is now easy to solve by noticing that for $\sigma_{k+1} \leq t \leq \sigma_k$, $k = 1, \ldots, r$, the optimal $s$ is

$$s = (\overbrace{t, \cdots, t}^{k \text{ terms}}, \sigma_{k+1}, \sigma_{k+2}, \cdots, \sigma_r),$$

and the loss for such $t$ is $\sum_{j=1}^k (\sigma_j - t)^2 + \lambda t$.

## A.2. Proof of Theorem 3.1

We argue by strong duality using a lemma of Theorem 1 in (Blanchet & Murthy, 2019).

**Lemma A.2.** *For $\gamma \geq 0$ and loss functions $l(x, \boldsymbol{B})$ that are upper semi-continuous in $x$ for each $\boldsymbol{B}$, define:*

$$\phi_\gamma(x_t; \boldsymbol{B}) := \sup_{u \in \mathbb{R}^d} \{l(u; \boldsymbol{B}) - \gamma c(u, x_t)\}. \tag{13}$$

*Then,*

$$\sup_{\mathbb{P}:\mathcal{D}_c(\mathbb{P},\mathbb{P}_n)\leq\delta} \mathbb{E}_{\mathbb{P}}[l(X; \boldsymbol{B})] = \min_{\gamma\geq 0}\left\{\gamma\delta + \frac{1}{n}\sum_{t=1}^{n}\phi_\gamma(x_t; \boldsymbol{B})\right\}. \tag{14}$$

Recall that our loss function is the total squared error: $l(X, \boldsymbol{B}) = \left\|X - \boldsymbol{B}^\top X\right\|_2^2 = X^\top \boldsymbol{H}\boldsymbol{H}^\top X$ where $\boldsymbol{H} := \boldsymbol{I} - \boldsymbol{B}$, and the cost function is $c(u, w) = \|w - u\|_2^2$. Using Lemma A.2, we can reduce the inner supremum of (7) to

$$\sup_{\mathbb{P}:\mathcal{D}_c(\mathbb{P},\mathbb{P}_n)\leq\delta} \mathbb{E}_{\mathbb{P}}[l(X; \boldsymbol{B})] = \min_{\gamma\geq 0}\{\gamma\delta + \frac{1}{n}\sum_{t=1}^{n}\phi_\gamma(x_t; \boldsymbol{B})\}, \tag{15}$$

where

$$\phi_\gamma(x_t; \boldsymbol{B}) := \sup_{u \in \mathbb{R}^d} \{l(u; \boldsymbol{B}) - \gamma c(u, x_t)\}$$
$$= \sup_{u \in \mathbb{R}^d} \{u^\top \boldsymbol{H}\boldsymbol{H}^\top u - \gamma \|x_t - u\|_2^2\}.$$

Rewriting $\Delta := u - x_t$, we have:

$$\phi_\gamma(x_t; \boldsymbol{B}) := \sup_{\Delta \in \mathbb{R}^d} \{(\Delta + x_t)^\top \boldsymbol{H}\boldsymbol{H}^\top(\Delta + x_t) - \gamma \|\Delta\|_2^2\}$$
$$= x_t^\top \boldsymbol{H}\boldsymbol{H}^\top x_t$$
$$+ \sup_{\Delta \in \mathbb{R}^d} \{\Delta^\top \boldsymbol{H}\boldsymbol{H}^\top \Delta + 2x_t^\top \boldsymbol{H}\boldsymbol{H}^\top \Delta - \gamma \|\Delta\|_2^2\}$$
$$= x_t^\top \boldsymbol{H}\boldsymbol{H}^\top x_t$$
$$+ \sup_{\Delta \in \mathbb{R}^d} \{-\Delta^\top(\gamma\boldsymbol{I} - \boldsymbol{H}\boldsymbol{H}^\top)\Delta + 2x_t^\top \boldsymbol{H}\boldsymbol{H}^\top \Delta\}.$$

Observe that inside the supremum is a quadratic function of $\Delta$, so the supreme is only finite if the quadratic function is concave. This means that $\gamma\boldsymbol{I} - \boldsymbol{H}\boldsymbol{H}^\top$ needs to be positive definite, and thus invertible, which requires that $\gamma > \lambda_1$ where $\lambda_1$ is the largest eigenvalue of $(\boldsymbol{H}\boldsymbol{H}^\top)$. Then, according to the first order condition, the supremum is achieved when $(\gamma\boldsymbol{I} - \boldsymbol{H}\boldsymbol{H}^\top)\Delta = \boldsymbol{H}\boldsymbol{H}^\top x_t$, i.e., $\Delta = (\gamma\boldsymbol{I} - \boldsymbol{H}\boldsymbol{H}^\top)^{-1}\boldsymbol{H}\boldsymbol{H}^\top x_t$. Plugging in the value for $\Delta$, we have

$$\phi_\gamma(x_t; \boldsymbol{B}) = x_t^\top \boldsymbol{H}\boldsymbol{H}^\top x_t + x_t^\top \boldsymbol{H}\boldsymbol{H}^\top(\gamma\boldsymbol{I} - \boldsymbol{H}\boldsymbol{H}^\top)^{-1}\boldsymbol{H}\boldsymbol{H}^\top x_t.$$

Through eigendecomposition, we can write

$$(\gamma\boldsymbol{I} - \boldsymbol{H}\boldsymbol{H}^\top)^{-1} = \boldsymbol{Q}(\gamma\boldsymbol{I} - \boldsymbol{\Lambda})^{-1}\boldsymbol{Q}^\top,$$

where $\boldsymbol{\Lambda}$ is the diagonal matrix with $\Lambda_{ii} = \lambda_i$ being the $i$-th largest eigenvalue of $\boldsymbol{H}\boldsymbol{H}^\top$, and $\boldsymbol{Q} = [Q_1 \, Q_2 \, \ldots \, Q_d]$ is the matrix whose $i$-th column is the eigenvector $Q_i$ of $\boldsymbol{H}\boldsymbol{H}^\top$ corresponding to the eigenvalue $\lambda_i$. Then

$$\phi_\gamma(x_t; \boldsymbol{B}) = x_t^\top \boldsymbol{H}\boldsymbol{H}^\top x_t + x_t^\top \boldsymbol{H}\boldsymbol{H}^\top \boldsymbol{Q}(\gamma\boldsymbol{I} - \boldsymbol{\Lambda})^{-1}\boldsymbol{Q}^\top \boldsymbol{H}\boldsymbol{H}^\top x_t$$
$$= x_t^\top \boldsymbol{Q}\boldsymbol{\Lambda}\boldsymbol{Q}^\top x_t + x_t^\top \boldsymbol{Q}\boldsymbol{\Lambda}(\gamma\boldsymbol{I} - \boldsymbol{\Lambda})^{-1}\boldsymbol{\Lambda}\boldsymbol{Q}^\top x_t$$

$$= x_t^\top Q \begin{bmatrix} \dfrac{\gamma\lambda_1}{\gamma - \lambda_1} & 0 & \cdots & 0 \\ 0 & \dfrac{\gamma\lambda_2}{\gamma - \lambda_2} & \cdots & 0 \\ \vdots & \vdots & \ddots & \vdots \\ 0 & 0 & \cdots & \dfrac{\gamma\lambda_d}{\gamma - \lambda_d} \end{bmatrix} Q^\top x_t$$

$$= \sum_{i=1}^{d} \frac{\gamma\lambda_i}{\gamma - \lambda_i}(Q_i^\top x_t)^2.$$

Now the minimum in (15) becomes:

$$\min_{\gamma > \lambda_1} \left\{ \gamma\delta + \frac{1}{n}\sum_{i=1}^{d} \frac{\gamma\lambda_i \sum_{t=1}^{n}(Q_i^\top x_t)^2}{\gamma - \lambda_i} \right\}. \tag{16}$$

Inside the minimum of (16) is a convex function of $\gamma$ on $\gamma > \lambda_1$ that tends to infinity as $\gamma \to \infty$ or $\gamma \to \lambda_1$. The optimal $\gamma$ should follow the first-order condition

$$\delta - \frac{1}{n}\sum_{i=1}^{d} \left( \frac{\lambda_i^2}{(\gamma - \lambda_i)^2} \sum_{t=1}^{n}(Q_i^\top x_t)^2 \right) = 0. \tag{17}$$

It is easy to see that this equation has a solution on $(\lambda_1, \infty)$, because the left hand side goes to $-\infty$ as $\gamma$ approaches $\lambda_1$, and goes to $\delta > 0$ as $\gamma$ approaches $\infty$. However, analytically solving this equation involves the $2d$-th order product $\prod_{i=1}^{d}(\gamma - \lambda_i)^2$ and is therefore difficult. So we introduce an approximation by replacing $(\gamma - \lambda_i)^2$ with $(\gamma - \lambda_1)^2$ in the denominator and replacing one of the $\lambda_i$'s in the numerator with $\lambda_1$[2]. In other words, we try to find the $\gamma$ that satisfies:

$$\delta - \frac{1}{n}\sum_{i=1}^{d} \left( \frac{\lambda_1\lambda_i}{(\gamma - \lambda_1)^2} \sum_{t=1}^{n}(Q_i^\top x_t)^2 \right) = 0, \tag{18}$$

which yields

$$\gamma = \lambda_1 + \frac{1}{\sqrt{\delta}}\sqrt{\lambda_1}\sqrt{\frac{1}{n}\sum_{t=1}^{n}\sum_{i=1}^{d}\lambda_i(Q_i^\top x_t)^2}$$

$$= \lambda_1 + \frac{1}{\sqrt{\delta}}\sqrt{\lambda_1}\sqrt{\frac{1}{n}\sum_{t=1}^{n}x_t^\top H H^\top x_t}.$$

---

[2]The informed reader might have noticed that the rest of the proof still follows if, instead of replacing with $\lambda_1$, we replace $\lambda_i$ with any number larger than or equal to $\lambda_1$. We choose $\lambda_1$ because it offers the tightest approximation of this simple form.

Using this value for $\gamma$, we obtain an upper bound on (16):

$$
\min_{\gamma > \lambda_1} \left\{ \gamma \delta + \frac{1}{n} \sum_{i=1}^{d} \frac{\gamma \lambda_i \sum_{t=1}^{n} (Q_i^\top x_t)^2}{\gamma - \lambda_i} \right\}
$$

$$
\leq \min_{\gamma > \lambda_1} \left\{ \gamma \delta + \frac{1}{n} \sum_{i=1}^{d} \frac{\gamma \lambda_i \sum_{t=1}^{n} (Q_i^\top x_t)^2}{\gamma - \lambda_1} \right\} \leq \min_{\gamma > \lambda_1} \left\{ \gamma \delta + \frac{\gamma \frac{1}{n} \sum_{t=1}^{n} x_t^\top H H^\top x_t}{\gamma - \lambda_1} \right\}
$$

$$
= \lambda_1 \delta + \sqrt{\delta} \sqrt{\lambda_1} \sqrt{\frac{1}{n} \sum_{t=1}^{n} x_t^\top H H^\top x_t} + \frac{\left( \lambda_1 + \frac{\sqrt{\lambda_1}}{\sqrt{\delta}} \sqrt{\frac{1}{n} \sum_{t=1}^{n} x_t^\top H H^\top x_t} \right) \left( \frac{1}{n} \sum_{t=1}^{n} x_t^\top H H^\top x_t \right)}{\frac{\sqrt{\lambda_1}}{\sqrt{\delta}} \sqrt{\frac{1}{n} \sum_{t=1}^{n} x_t^\top H H^\top x_t}}
$$

$$
= \lambda_1 \delta + 2 \sqrt{\delta \lambda_1 \frac{1}{n} \sum_{t=1}^{n} x_t^\top H H^\top x_t} + \frac{1}{n} \sum_{t=1}^{n} x_t^\top H H^\top x_t
$$

$$
= \left( \sqrt{\frac{1}{n} \sum_{t=1}^{n} x_t^\top H H^\top x_t} + \sqrt{\delta} \sqrt{\lambda_1} \right)^2 = \left( \sqrt{\frac{1}{n} \sum_{t=1}^{n} x_t^\top H H^\top x_t} + \sqrt{\delta} \| H \|_2 \right)^2,
$$

where $\| H \|_2$ is the spectral norm of $H$ and is equal to its largest singular value $\sqrt{\lambda_1}$.

Finally, we derive a lower bound on (16) to show the tightness of our relaxation. Since

$$
\frac{\gamma}{\gamma - \lambda_i} \geq 1, \qquad \forall i \in [d], \ \gamma > \lambda_1,
$$

we have

$$
\min_{\gamma > \lambda_1} \left\{ \gamma \delta + \frac{1}{n} \sum_{i=1}^{d} \frac{\gamma \lambda_i \sum_{t=1}^{n} (Q_i^\top x_t)^2}{\gamma - \lambda_i} \right\} \geq \min_{\gamma > \lambda_1} \left\{ \gamma \delta + \frac{1}{n} \sum_{i=1}^{d} \lambda_i \sum_{t=1}^{n} (Q_i^\top x_t)^2 \right\}
$$

$$
= \min_{\gamma > \lambda_1} \left\{ \gamma \delta + \frac{1}{n} \sum_{t=1}^{n} x_t^\top H H^\top x_t \right\} = \min_{\gamma > \lambda_1} \left\{ \gamma \delta + \frac{1}{n} \sum_{t=1}^{n} x_t^\top H H^\top x_t \right\}
$$

$$
> \lambda_1 \delta + \frac{1}{n} \sum_{t=1}^{n} x_t^\top H H^\top x_t \geq \frac{1}{2} \left( \sqrt{\delta \lambda_1} + \sqrt{\frac{1}{n} \sum_{t=1}^{n} x_t^\top H H^\top x_t} \right)^2.
$$

The last inequality follows from the elementary fact $a^2 + b^2 \geq (a+b)^2/2, \forall a, b \geq 0$.

$\square$

## A.3. Proof of Theorem B.4

By Proposition 3 in (Blanchet et al., 2019), we have for any $\boldsymbol{H} \in \mathbb{R}^{d \times d}$ and $\mathrm{diag}(\boldsymbol{H}) = 1$,

$$
\begin{aligned}
\mathcal{R}_n(\boldsymbol{H}) = \sup_{\boldsymbol{\Lambda} \in \mathbb{R}^{d \times d}} &\left\{ -\mathbb{E}_{\mathbb{P}_n} \left[ \sup_{u \in \mathbb{R}^d} \left\{ \sum_{i,j \in [d], i \neq j} \lambda_{ij} \left( uu^\top \boldsymbol{H} \right)_{ij} \right. \right. \right. \\
&\left. \left. \left. - \|u - X\|_2^2 \right\} \right] \right\} \\
= \sup_{\boldsymbol{\Lambda} \in \mathbb{R}^{d \times d}: \mathrm{diag}(\boldsymbol{\Lambda}) = 0} &\left\{ -\mathbb{E}_{\mathbb{P}_n} \left[ \sup_{u \in \mathbb{R}^d} \left\{ \mathrm{tr}(\boldsymbol{\Lambda}^\top uu^\top \boldsymbol{H}) \right. \right. \right. \\
&\left. \left. \left. - \|u - X\|_2^2 \right\} \right] \right\}
\end{aligned}
$$

Define $h(X, \boldsymbol{H}) := XX^\top \boldsymbol{H}$, then observe that the inner-most supremum

$$
\begin{aligned}
&\sup_{u \in \mathbb{R}^d} \left\{ \mathrm{tr}(\boldsymbol{\Lambda}^\top uu^\top \boldsymbol{H}) - \|u - X\|_2^2 \right\} \\
&= \sup_{\Delta \in \mathbb{R}^d} \left\{ \mathrm{tr}(\boldsymbol{\Lambda}^\top h(X + \Delta, \boldsymbol{H})) - \|\Delta\|_2^2 \right\} \\
&= \sup_{\Delta \in \mathbb{R}^d} \left\{ \mathrm{tr}(\boldsymbol{\Lambda}^\top [h(X + \Delta, \boldsymbol{H}) - h(X, \boldsymbol{H})]) - \|\Delta\|_2^2 \right\} \\
&\quad + \mathrm{tr}(\boldsymbol{\Lambda}^\top h(X, \boldsymbol{H})).
\end{aligned}
$$

We can write

$$
\begin{aligned}
&\mathrm{tr}(\boldsymbol{\Lambda}^\top [h(X + \Delta, \boldsymbol{H}) - h(X, \boldsymbol{H})]) \\
&= \int_0^1 \frac{d}{dt} \mathrm{tr}(\boldsymbol{\Lambda}^\top h(X + t\Delta, \boldsymbol{H})) dt.
\end{aligned}
$$

Calculating the derivative, we have

$$
\begin{aligned}
\frac{d}{dt} \mathrm{tr}(\boldsymbol{\Lambda}^\top h(X + t\Delta, \boldsymbol{H})) &= 2 \mathrm{tr}(\boldsymbol{H} \boldsymbol{\Lambda}^\top (X + t\Delta)\Delta^\top) \\
&= 2 \mathrm{tr}(\boldsymbol{H} \boldsymbol{\Lambda}^\top X \Delta^\top) + 2t\Delta^\top \boldsymbol{H} \boldsymbol{\Lambda}^\top \Delta,
\end{aligned}
$$

which is linear in $t$. So we deduce

$$
\begin{aligned}
\mathcal{R}_n(\boldsymbol{H}) = \sup_{\boldsymbol{\Lambda} \in \mathbb{R}^{d \times d}: \mathrm{diag}(\boldsymbol{\Lambda}) = 0} &\left\{ -\mathbb{E}_{\mathbb{P}_n} \left[ \phantom{\sup} \right. \right. \\
&\sup_{\Delta \in \mathbb{R}^d} \left\{ 2 \mathrm{tr}(\boldsymbol{H} \boldsymbol{\Lambda}^\top X \Delta^\top) \right. \\
&\left. + \Delta^\top \boldsymbol{H} \boldsymbol{\Lambda}^\top \Delta - \|\Delta\|_2^2 \right\} \\
&\left. \left. + \mathrm{tr}(\boldsymbol{\Lambda}^\top XX^\top \boldsymbol{H}) \right] \right\}.
\end{aligned}
$$

Introduce the scaling $\Delta = \bar{\Delta}/n^{1/2}$ and $\bar{\Lambda} = \Lambda n^{1/2}$. Then we have

$$
nR_n(\boldsymbol{H}) = \sup_{\bar{\Lambda} \in \mathbb{R}^{d \times d}: \mathrm{diag}(\bar{\Lambda})=0} \Bigg\{ - \mathbb{E}_{\mathbb{P}_n} \Bigg[
$$
$$
\sup_{\bar{\Delta} \in \mathbb{R}^d} \Big\{ 2 \operatorname{tr}(\boldsymbol{H}\bar{\Lambda}^\top X \bar{\Delta}^\top)
$$
$$
+ \bar{\Delta}^\top \boldsymbol{H} \bar{\Lambda}^\top \bar{\Delta}/n^{1/2} - \big\| \bar{\Delta} \big\|_2^2 \Big\}
$$
$$
+ n^{1/2} \operatorname{tr}(\bar{\Lambda}^\top X X^\top \boldsymbol{H}) \Bigg] \Bigg\}.
$$

Under Assumption B.3, we have, for any matrix $\Lambda \in \mathbb{R}^{d \times d}$ such that $\mathrm{diag}(\Lambda) = 0$ and $\Lambda \neq \boldsymbol{0}$,

$$
\mathbb{P}^* \left( \sum_{i=1}^d \left( \operatorname{tr}(X h_{i\cdot}^{*\top} \boldsymbol{\Lambda}^\top + X \lambda_{i\cdot}^\top \boldsymbol{H}^{*\top}) \right)^2 > 0 \right) > 0,
$$

where $h_{i\cdot}^{*\top}$ represents the $i$-th row of $\boldsymbol{H}^*$ and $\lambda_{i\cdot}^\top$ the $i$-th row of $\boldsymbol{\Lambda}$. Then, Assumptions A2) - A4) in (Blanchet et al., 2019) are satisfied, and by Lemma 2 in (Blanchet et al., 2019), for every $\varepsilon > 0$, there exists $n_0 > 0$ and $b \in (0, \infty)$ such that for all $n \geq n_0$,

$$
\mathbb{P}\Bigg( \sup_{\|\bar{\Lambda}\|_F \geq b} \Bigg\{ - \mathbb{E}_{\mathbb{P}_n} \Bigg[ \sup_{\bar{\Delta} \in \mathbb{R}^d} \Big\{ 2 \operatorname{tr}(\boldsymbol{H}\bar{\Lambda}^\top X \bar{\Delta}^\top)
$$
$$
+ \bar{\Delta}^\top \boldsymbol{H} \bar{\Lambda}^\top \bar{\Delta}/n^{1/2} - \big\| \bar{\Delta} \big\|_2^2 \Big\}
$$
$$
+ n^{1/2} \operatorname{tr}(\bar{\Lambda}^\top X X^\top \boldsymbol{H}) \Bigg] \Bigg\} > 0 \Bigg) \leq \varepsilon.
$$

This result means that if we want the value in the outer supremum to be larger than 0 with high probability as $n$ approaches infinity, we need $\|\bar{\Lambda}\|_F$ smaller than a finite $b$. In other words, the $\bar{\Lambda}^*$ that attains the supremum will have $\|\bar{\Lambda}^*\|_F$ smaller than a finite $b$. In this case, for any fixed $\boldsymbol{H}$, $\|\boldsymbol{H}\|_F$ should be finite, then

$$
\bar{\Delta}^\top \boldsymbol{H} \bar{\Lambda}^{*\top} \bar{\Delta}/n^{1/2} \leq \big\| \bar{\Delta} \big\|_2^2 \|\boldsymbol{H}\|_F \big\| \bar{\Lambda}^* \big\|_F /n^{1/2}
$$
$$
\leq b \big\| \bar{\Delta} \big\|_2^2 \|\boldsymbol{H}\|_F /n^{1/2},
$$

which is negligible compared with $\big\| \bar{\Delta} \big\|_2^2$ as $n \to \infty$. The remaining terms in the inner supremum can be simplified:

$$
\sup_{\bar{\Delta} \in \mathbb{R}^d} \Big\{ 2 \operatorname{tr}(\boldsymbol{H}\bar{\Lambda}^\top X \bar{\Delta}^\top) - \big\| \bar{\Delta} \big\|_2^2 \Big\}
$$
$$
= \sup_{\bar{\Delta} \in \mathbb{R}^d} \Big\{ 2 \big\| \boldsymbol{H}\bar{\Lambda}^\top X \big\|_2 \big\| \bar{\Delta} \big\|_2 - \big\| \bar{\Delta} \big\|_2^2 \Big\}
$$
$$
= \big\| \boldsymbol{H}\bar{\Lambda}^\top X \big\|_2^2.
$$

Also, we can write

$$
\mathbb{E}_{\mathbb{P}_n} \Big[ n^{1/2} \operatorname{tr}(\bar{\Lambda}^\top X X^\top \boldsymbol{H}) \Big] = \operatorname{tr}\Big( n^{1/2}\big( \bar{\Lambda}^\top \mathbb{E}_{\mathbb{P}_n} \big[ X X^\top \big] \boldsymbol{H}
$$
$$
- \bar{\Lambda}^\top \mathbb{E}_{\mathbb{P}^*} \big[ X X^\top \big] \boldsymbol{H} \big) \Big)
$$

because the diagonals of $\bar{\Lambda}^\top$ are zero, and by definition, the off-diagonals of $\mathbb{E}_{\mathbb{P}^*} \big[ X X^\top \big] \boldsymbol{H}$ are zero, thus the additional term $\bar{\Lambda}^\top \mathbb{E}_{\mathbb{P}^*} \big[ X X^\top \big] \boldsymbol{H} = \boldsymbol{0}$. Then by Assumption B.1, as $n \to \infty$,

$$
\mathbb{E}_{\mathbb{P}_n} \Big[ n^{1/2} \operatorname{tr}(\bar{\Lambda}^\top X X^\top \boldsymbol{H}) \Big] \implies \operatorname{tr}\Big( \bar{\Lambda}^\top \boldsymbol{Z} \boldsymbol{H} \Big)
$$

where $\boldsymbol{Z} \sim N(0, \Upsilon_g)$, and $g(X) := XX^\top$. Finally, as $n \to \infty$,

$$\mathbb{E}_{\mathbb{P}_n}\left[\left\|\boldsymbol{H}\bar{\boldsymbol{\Lambda}}^\top X\right\|_2^2\right] \Rightarrow \mathbb{E}_{\mathbb{P}^*}\left[\left\|\boldsymbol{H}\bar{\boldsymbol{\Lambda}}^\top X\right\|_2^2\right].$$

Because the dimension of $\bar{\boldsymbol{\Lambda}}$ is fixed at $d$, we can safely take the limit inside the supremum. Therefore, we conclude that, as $n \to \infty$,

$$n\mathcal{R}_n(\boldsymbol{H}) \Rightarrow$$

$$\sup_{\bar{\boldsymbol{\Lambda}}\in\mathbb{R}^{d\times d}:\mathrm{diag}(\bar{\boldsymbol{\Lambda}})=0}\left\{-\mathbb{E}_{\mathbb{P}^*}\left[\left\|\boldsymbol{H}\bar{\boldsymbol{\Lambda}}^\top X\right\|_2^2\right] - \mathrm{tr}\left(\bar{\boldsymbol{\Lambda}}^\top \boldsymbol{Z}\boldsymbol{H}\right)\right\}.$$

This supremum can be bounded from above by substituting $\boldsymbol{H}\boldsymbol{\Lambda}^\top$ with any $\boldsymbol{G} \in \mathbb{R}^{d\times d}$:

$$\sup_{\bar{\boldsymbol{\Lambda}}\in\mathbb{R}^{d\times d}:\mathrm{diag}(\bar{\boldsymbol{\Lambda}})=0}\left\{-\mathbb{E}_{\mathbb{P}^*}[\left\|\boldsymbol{H}\bar{\boldsymbol{\Lambda}}^\top X\right\|_2^2] - \mathrm{tr}(\bar{\boldsymbol{\Lambda}}^\top \boldsymbol{Z}\boldsymbol{H})\right\}$$
$$\leq \sup_{\boldsymbol{G}\in\mathbb{R}^{d\times d}}\left\{-\mathbb{E}_{\mathbb{P}^*}[\|\boldsymbol{G}X\|_2^2] - \mathrm{tr}(\boldsymbol{G}\boldsymbol{Z})\right\}.$$

Breaking up $\boldsymbol{G}$ into rows, where the $i$-th *row* is $G_{i\cdot}$, and let $Z_{\cdot i}$ be the $i$-th *column* of $\boldsymbol{Z}$, we have

$$n\mathcal{R}_n(\boldsymbol{H})$$
$$\lesssim_D \sup_{\bar{\boldsymbol{\Lambda}}\in\mathbb{R}^{d\times d}:\mathrm{diag}(\bar{\boldsymbol{\Lambda}})=0}\left\{\sum_{i=1}^d\left(-\mathbb{E}_{\mathbb{P}^*}[(G_{i\cdot}^\top X)^2] - G_{i\cdot}^\top Z_{\cdot i}\right)\right\}.$$

Taking the derivative with respect to $G_{i\cdot}$, we obtain

$$-2\mathbb{E}_{\mathbb{P}^*}[X^\top G_{i\cdot}X] - Z_{\cdot i} = 0. \tag{19}$$

Let $\boldsymbol{\Sigma}_* = \mathbb{E}_{\mathbb{P}^*}(XX^\top)$, which we assume to be invertible. Then (19) can be written as

$$-2\boldsymbol{\Sigma}_* G_{i\cdot} - Z_{\cdot i} = 0,$$

which has a unique solution:

$$G_{i\cdot} = -\frac{1}{2}\boldsymbol{\Sigma}_*^{-1}Z_{\cdot i},$$

where $\boldsymbol{\Sigma}_*^{-1}$ is the inverse of $\boldsymbol{\Sigma}_*$. Therefore,

$$\begin{aligned} & -\mathbb{E}_{\mathbb{P}^*}[(G_{i\cdot}^\top X)^2] - G_{i\cdot}^\top Z_{\cdot i} \\ =& -G_{i\cdot}^\top \boldsymbol{\Sigma}_* G_{i\cdot} - G_{i\cdot}^\top Z_{\cdot i} \\ =& -\frac{1}{4}Z_{\cdot i}^\top\boldsymbol{\Sigma}_*^{-1}\boldsymbol{\Sigma}_*\boldsymbol{\Sigma}_*^{-1}Z_{\cdot i} + \frac{1}{2}Z_{\cdot i}^\top\boldsymbol{\Sigma}_*^{-1}Z_{\cdot i} \\ =& \frac{1}{4}Z_{\cdot i}^\top\boldsymbol{\Sigma}_*^{-1}Z_{\cdot i}, \end{aligned}$$

and we conclude that

$$n\mathcal{R}_n(\boldsymbol{H}^*) \lesssim_D \sum_{i=1}^d \frac{1}{4}Z_{\cdot i}^\top\boldsymbol{\Sigma}_*^{-1}Z_{\cdot i},$$

where $Z_{\cdot i}$ is the $i$-th column of $\boldsymbol{Z} \sim N(0, \Upsilon_g)$. $\qquad\square$

# B. Choice of $\delta$ for DRO

The strength of the regularization, controlled by $\delta$, is usually determined exogenously or by cross-validation in the machine learning literature. However, since $\delta$ is the radius of the uncertainty region in our setting, the choice of $\delta$ should be informed by the degree of uncertainty in the data. Specifically, we determine a distributional uncertainty region in a way that it is just large enough so that the correct set of regression coefficients, which we would obtain if the true distribution were known, becomes a plausible choice with a sufficiently high confidence level. A simple, actionable recipe for choosing $\delta$ is provided at the end of this subsection.

Define the covariance of a random matrix $\boldsymbol{M} \in \mathbb{R}^{d \times d}$, denoted by $\mathrm{Cov}(\boldsymbol{M})$, as a $(d \times d) \times (d \times d)$ tensor, with $\mathrm{Cov}(\boldsymbol{M})_{ij,kl} := \mathrm{Cov}(M_{ij}, M_{kl})$, $i, j, k, l \in [d]$. Before describing our method for choosing $\delta$, we introduce the following assumptions:

**Assumption B.1.** The time series $\{X(t) \in \mathbb{R}^d : t \geq 0\}$ underlying the observations is a stationary, ergodic process satisfying $\mathbb{E}_{\mathbb{P}^*}\left(\left\|X(t)\right\|_2^4\right) < \infty$ for each $t \geq 0$. Moreover, for each measurable function $g : \mathbb{R}^d \to \mathbb{R}^{d \times d}$ such that $\sum_{i,j} |g(x)_{ij}| \leq c(1 + \|x\|_2^2)$ for some $c > 0$, the limit

$$\Upsilon_g := \lim_{n \to \infty} \mathrm{Cov}_{\mathbb{P}^*}\left(n^{-1/2} \sum_{t=1}^n g(X(t))\right) \in \mathbb{R}^{(d \times d) \times (d \times d)}$$

exists, and the central limit theorem holds:

$$n^{1/2}\left[\mathbb{E}_{\mathbb{P}_n}\left(g(X)\right) - \mathbb{E}_{\mathbb{P}^*}\left(g(X)\right)\right] \Rightarrow N(0, \Upsilon_g),$$

where "$\Rightarrow$" denotes weak convergence as $n \to \infty$ with fixed $d$, and $N(0, \Upsilon_g)$ represents a random matrix $\boldsymbol{Z}$ whose entries follow a normal distribution with $\mathbb{E}[Z_{ij}] = 0$ and $\mathrm{Cov}(Z_{ij}, Z_{kl}) = (\Upsilon_g)_{ij,kl}$.

**Assumption B.2.** The classical optimization problem (6) has a unique solution $\boldsymbol{B}^*$.

**Assumption B.3.** $X(t)$ has a density for each $t \geq 0$.

Assumption B.1 is standard for most time series models. Assumption B.2 holds when the true underlying covariance matrix is invertible, which is true when no random variable is exactly a linear combination of other random variables. This condition is easily satisfied when, for example, each random variable is generated with an idiosyncratic noise.

In order to choose an appropriate $\delta$, we follow the idea behind the robust Wasserstein profile inference (RWPI) approach introduced in (Blanchet et al., 2019). Intuitively, the uncertainty region $\mathcal{U}_\delta(\mathbb{P}_n) := \{\mathbb{P} : \mathcal{D}_c(\mathbb{P}, \mathbb{P}_n) \leq \delta\}$ contains all the probability measures that are plausible variations of $\mathbb{P}_n$ implied by the data. Let $\boldsymbol{H} := \boldsymbol{I} - \boldsymbol{B}$ for simpler notation. We denote by $\mathcal{Q}(\mathbb{P})$ the classical regression problem with $\mathbb{P}$ being the underlying probability distribution:

$$\underset{\boldsymbol{H} \in \mathbb{R}^{d \times d}}{\mathrm{minimize}} \quad \mathbb{E}_{\mathbb{P}}\left[X^\top \boldsymbol{H} \boldsymbol{H}^\top X\right], \quad \text{s.t.} \quad \mathrm{diag}(\boldsymbol{H}) = 1.$$

Also, denote by $\boldsymbol{H}_\mathbb{P}$ a solution to $\mathcal{Q}(\mathbb{P})$ and by $\mathcal{H}_\mathbb{P}$ the set of all such solutions. According to Assumption B.2, we have $\mathcal{H}_{\mathbb{P}^*} = \{\boldsymbol{H}^*\}$ for some $\boldsymbol{H}^* := \boldsymbol{I} - \boldsymbol{B}^*$. Therefore, there exist unique Lagrange multipliers $\lambda_1^*, \lambda_2^*, \ldots, \lambda_d^*$ such that

$$\mathbb{E}_{\mathbb{P}^*}\left[XX^\top\right]\boldsymbol{H}^* - \boldsymbol{\Lambda}^* = \boldsymbol{0}, \quad \mathrm{diag}(\boldsymbol{H}^*) = 1,$$

where $\boldsymbol{\Lambda}^*$ is the diagonal matrix with entries $\lambda_1^*, \lambda_2^*, \ldots, \lambda_d^*$.

We choose $\delta > 0$ such that $\mathcal{U}_\delta(\mathbb{P}_n)$ contains all probability distributions that are plausible variations of $\mathbb{P}_n$, and hence $\boldsymbol{H}_\mathbb{P}$ with $\mathbb{P} \in \mathcal{U}_\delta(\mathbb{P}_n)$ is a plausible estimate of $\boldsymbol{H}^*$. Thus, if we collect all such plausible estimates as the set:

$$\Lambda_\delta(\mathbb{P}_n) = \bigcup_{\mathbb{P} \in \mathcal{U}_\delta(\mathbb{P}_n)} \mathcal{H}_\mathbb{P},$$

then $\Lambda_\delta(\mathbb{P}_n)$ is a natural confidence region for $\boldsymbol{H}^*$. Therefore, $\delta$ should be chosen as the smallest number $\delta_n^*$ such that $\boldsymbol{H}^*$ falls in this region with a given confidence level:

$$\delta_n^* = \min\left\{\delta : \mathbb{P}^*\left(\boldsymbol{H}^* \in \Lambda_\delta(\mathbb{P}_n)\right) \geq 1 - \alpha\right\},$$

where $1 - \alpha$ is a user-defined confidence level (typically 95%).

In order to be able to compute $\delta_n^*$, we provide a simpler representation using an auxiliary function called the Robust Wasserstein Profile (RWP) function. First observe that any $\boldsymbol{H} \in \Lambda_\delta(\mathbb{P}_n)$ if and only if there exist $\mathbb{P} \in \mathcal{U}_\delta(\mathbb{P}_n)$ along with $\lambda_1, \lambda_2, \ldots, \lambda_d \in (-\infty, \infty)$ and their corresponding diagonal matrix $\boldsymbol{\Lambda}$ such that

$$\mathbb{E}_{\mathbb{P}} \left[ XX^\top \right] \boldsymbol{H} - \boldsymbol{\Lambda} = \boldsymbol{0}, \quad \text{diag}(\boldsymbol{H}) = 1.$$

By plugging the second equation into the first, we have

$$\lambda_i = - \left( \mathbb{E}_{\mathbb{P}} \left[ XX^\top \right] \boldsymbol{H} \right)_{ii} - \mathbb{E}_{\mathbb{P}} \left[ X_i^2 \right] (1 - h_{ii}), \text{ for } i \in [d],$$

where $h_{ii}$ is the $i$-th element of the $i$-th row of $\boldsymbol{H}$. Then the system of equations that $\boldsymbol{H}$ needs to satisfy becomes:

$$1 - h_{ii} = 0 \text{ and } \left( \mathbb{E}_{\mathbb{P}} \left[ XX^\top \right] \boldsymbol{H} \right)_{ij} = 0, \quad \forall i, j \in [d] \text{ and } i \neq j.$$

Now we define the following RWP function

$$\mathcal{R}_n(\boldsymbol{H}) := \inf \left\{ \mathcal{D}_c(\mathbb{P}, \mathbb{P}_n) : 1 - h_{ii} = 0, \left( \mathbb{E}_{\mathbb{P}} \left[ XX^\top \right] \boldsymbol{H} \right)_{ij} = 0, \text{ for } i, j \in [d] \text{ and } i \neq j \right\}$$

for $\boldsymbol{H} \in \mathbb{R}^{d \times d}$ where $\mathcal{S}_+^{d \times d}$. Then, we can rewrite $\delta_n^*$ as:

$$\delta_n^* = \inf \left\{ \delta : \mathbb{P}^* \left( \mathcal{R}_n(\boldsymbol{H}^*) \leq \delta \right) \geq 1 - \alpha \right\}. \tag{20}$$

In other words, $\delta_n^*$ is now the $1 - \alpha$ quantile of $\mathcal{R}_n(\boldsymbol{H}^*)$. If we can asymptotically approximate the distribution of $\mathcal{R}_n(\boldsymbol{H}^*)$, $\delta_n^*$ can then be easily determined.

Before presenting the asymptotic distribution of $\mathcal{R}_n(\boldsymbol{H}^*)$, we first introduce the notation for asymptotic stochastic upper bound $n\mathcal{R}_n(\boldsymbol{H}^*) \lesssim_D \bar{\mathcal{R}}$, which means that, for every continuous and bounded non-decreasing function $f(\cdot)$, we have

$$\limsup_{n \to \infty} \mathbb{E} \left[ f \left( n\mathcal{R}_n(\boldsymbol{H}^*) \right) \right] \leq \mathbb{E} \left[ f(\bar{\mathcal{R}}) \right].$$

Similarly, we write $\gtrsim_D$ for an asymptotic stochastic lower bound, namely

$$\liminf_{n \to \infty} \mathbb{E} \left[ f \left( n\mathcal{R}_n(\boldsymbol{H}^*) \right) \right] \geq \mathbb{E} \left[ f(\bar{\mathcal{R}}) \right].$$

If both the stochastic upper and lower bounds hold for the same $\bar{\mathcal{R}}$, then $n\mathcal{R}_n(\boldsymbol{H}^*) \Rightarrow \bar{\mathcal{R}}$.

Now let us state an asymptotic stochastic upper bound for $n\mathcal{R}_n(\boldsymbol{H}^*)$.

**Theorem B.4.** *Under Assumptions B.1 and B.3, write $\boldsymbol{\Sigma}_* := \mathbb{E}_{\mathbb{P}^*} \left[ XX^\top \right]$ and $g(X) := XX^\top$. If $\boldsymbol{\Sigma}_*$ is invertible, then*

$$n\mathcal{R}_n(\boldsymbol{H}^*) \lesssim_D \bar{\mathcal{R}} := \sum_{i=1}^d \frac{1}{4} Z_{\cdot i}^\top \boldsymbol{\Sigma}_*^{-1} Z_{\cdot i}$$

*where $Z_{\cdot i}$ is the $i$-th column of $\boldsymbol{Z} \sim N(0, \Upsilon_g)$.*

The result of Theorem B.4 involves $\boldsymbol{\Sigma}_*^{-1}$. The true covariance matrix $\boldsymbol{\Sigma}_*$ can be estimated using the sample second-moment matrix $\boldsymbol{\Sigma}_n = \mathbb{E}_{\mathbb{P}_n} \left[ XX^\top \right] = \frac{1}{n-1} \sum_{t=1}^n g(x_t)$. However, estimating $\boldsymbol{\Sigma}_*^{-1}$ with $\boldsymbol{\Sigma}_n^{-1}$ is not possible when $n < d$. Even when $n$ is moderately large but of the same order as $d$, the sample covariance matrix has been shown to be unreliable (e.g., (Johnstone, 2001)). Here, we apply a commonly used remedy in machine learning, i.e., estimating $\boldsymbol{\Sigma}_*^{-1}$ by only keeping the diagonals of $\boldsymbol{\Sigma}_*$ when calculating its inverse; see e.g., (Bickel & Levina, 2004). After this, we can obtain $\delta_n^*$ as the $1 - \alpha$ quantile of $\bar{\mathcal{R}}/n$, as long as we know the distribution of $\boldsymbol{Z}$. We can draw samples from the distribution of $\boldsymbol{Z}$ and then numerically estimate the quantile of $\bar{\mathcal{R}}$. $\boldsymbol{Z}$ follows a normal distribution with a covariance matrix $\Upsilon_g$, which can be estimated using the sample covariances of observations of $g(x_t) \in \mathbb{R}^{d \times d}$, $t = 1, \ldots, n$. We note that since $\boldsymbol{Z}$ is a random symmetric

matrix in $\mathbb{R}^{d \times d}$, the covariances of its entries $\Upsilon_g$ is a $(d \times d) \times (d \times d)$ tensor. Nonetheless, $(\Upsilon_g)_{ij,kl}$ which represents the covariance between $Z_{ij}$ and $Z_{kl}$ can be approximated by the sample covariance $\frac{1}{n-1} \sum_{t=1}^{n} \left( g(x_t)_{ij} - \bar{g}_{ij} \right) \left( g(x_t)_{kl} - \bar{g}_{kl} \right)$, where $\bar{g}_{ij} := \frac{1}{n} \sum_{t=1}^{n} g(x_t)_{ij}$. One should, however, be mindful that applying this method is not always realistic in practice. First of all, $\Upsilon_g$ has size $d^4$ and can be difficult to fit in the RAM of a consumer computer (e.g., when $d = 500$, $\Upsilon_g$ is roughly 250 GB in float32 format). Further, it would require $n > d^2$ observations for the sample covariance matrix to be positive definite. In many applications, the number of observations of $n$ is on the same order as $d$, so the $\Upsilon_g$ estimated this way could be highly unstable. An alternative method is to simply disregard the covariances assuming entries in $\boldsymbol{Z}$ are independent, and only calculate the diagonals. Recall that $\Upsilon_g = \lim_{n \to \infty} \mathrm{Cov}_{\mathbb{P}^*} \left( n^{-1/2} \sum_{t=1}^{n} g(X(t)) \right)$. Because $g(x) := xx^\top$, $\sum_{t=1}^{n} g(X(t))$ follows the Wishart distribution with degree of freedom $n$ if we further assume that $X$ is normal, and its variance is $n \left[ \sigma_{ii} \sigma_{jj} + (\sigma_{ij})^2 \right]$. Then the diagonals of $\Upsilon_g$ can be computed[3]: $(\Upsilon_g)_{ij,ij} = \sigma_{ii} \sigma_{jj} + (\sigma_{ij})^2$. The independence also greatly simplifies the sampling of $\boldsymbol{Z}$. We now provide a simple recipe for choosing $\delta$:

1. Collect standardized data $\{x_t\}_{t=1}^{n}$, $x_t \in \mathbb{R}^d$.

2. Calculate second moments $\left\{ g(x_t) = x_t x_t^\top \right\}_{t=1}^{n}$.

3. Use the sample second-moment matrix $\boldsymbol{\Sigma}_n = \mathbb{E}_{\mathbb{P}_n} \left[ XX^\top \right] = \frac{1}{n-1} \sum_{t=1}^{n} g(x_t)$ to approximate $\boldsymbol{\Sigma}_*$. Then estimate $\boldsymbol{\Sigma}_*^{-1}$ by only keeping the diagonals of $\boldsymbol{\Sigma}_*$.

4. Calculate $\Upsilon_g$ using either of the following methods:

   (a) $(\Upsilon_g)_{ij,kl} = \dfrac{1}{n-1} \sum_{t=1}^{n} \left( g(x_t)_{ij} - \bar{g}_{ij} \right) \left( g(x_t)_{kl} - \bar{g}_{kl} \right).$

   (b) $(\Upsilon_g)_{ij,ij} = \sigma_{ii} \sigma_{jj} + (\sigma_{ij})^2$, $(\Upsilon_g)_{ij,kl} = 0$ if $(k,l) \neq (i,j)$.

5. Draw $M$ samples $\{\boldsymbol{Z}_m\}_{m=1}^{M}$ from the distribution $N(0, \Upsilon_g)$ to numerically estimate the $1 - \alpha$ quantile of $\bar{\mathcal{R}}/n$. Apply Theorem B.4 and (20) to set $\delta = \delta_n^*$ to this quantile.

## C. Implementation details and additional results of the financial data experiment

### C.1. Clustering and portfolio construction

Method 1 is a combination of the DRO subspace clustering and the ACC clustering in a hierarchical fashion. We believe clusters generated by this approach is more suitable for stock selection, compared to, for instance, clusters generated directly by subspace clustering algorithms. This is because stocks in the same low-dimensional subspace may still be quite different from each other (vectors in the same subspace can point to rather different directions), and it may be difficult to use a single stock to represent a whole cluster. With the DRO clustering at the higher level followed by the ACC at the lower level, the former breaks down the universe into stocks driven by *groups* of factors, and the latter then easily finds stocks most closely associated with each single factor. We parsimoniously choose $K_1 = K_2$ since we have no prior knowledge of how many subspace there should be vs. the dimensions of these subspaces. We compare this method with Method 2, which directly applies the ACC algorithm on the full universe. The ACC algorithm works very well in this task as demonstrated in (Tang et al., 2022). We also include other methods as benchmarks.

At the end of the first trading day of each February, the above three clustering methods are applied to daily log returns in the backward 500-trading-day window for valid constituent stocks as described in Section 6.2. For each clustering method, we create a total of 19 sets of clusters, one for each year. The choice of parameters for the clustering algorithms is the same as described in Section 4.

### C.2. Details of portfolio construction

The volatility of a stock is measured by the sample variance of daily returns in the past $n = 500$ trading days, the same window used for clustering. From a practical and empirical perspective, the reason why we choose low volatility as the criterion is twofold. First, volatility as a criterion does not involve the estimation of the mean returns. All clustering

---

[3]The off-diagonals can also be computed: http://personal.psu.edu/drh20/asymp/fall2002/lectures/ln08.pdf

algorithms tested avoid using the stocks' mean returns. It would then be inconsistent if we selected stocks from the clusters based on return-related criteria, e.g., mean return or Sharpe ratio. More importantly, the estimation of mean returns is well known to be often inaccurate (the "mean-blur" problem; see, e.g., (Merton, 1980)), rendering return-related criteria unreliable of indicating future performance. The second reason is that stocks with low volatility have been observed to outperform the benchmarks over time, which is contrary to CAPM and is documented as the "low-risk anomaly" (e.g., (Zaremba & Shemer, 2017)). We only choose one stock with the lowest volatility from each cluster, yielding the same number of stocks as clusters for each clustering method every time we update the portfolio.

The minimum variance allocation strategy is similar to Markowitz's mean-variance optimization but without the expected return constraint:

$$\min \quad w^\top \boldsymbol{\Sigma} w$$
$$\text{s.t.} \quad w^\top 1 = 1, \quad w \geq 0.$$

We choose the minimum variance allocation because it also does not involve the estimation of the mean return. Similar to why we use low volatility as a criterion to select stocks from the clusters, we aim to keep the experiment consistent by avoiding the estimation of the mean returns throughout the experiment.

The portfolios are updated annually. At each portfolio update, a new set of stocks are selected according to the clustering result. Their allocations are calculated using all daily returns in the past 500 trading days, starting with the first day when all stocks are available. The positions are then held until the first trading day of the following update. Any dividends are immediately reinvested in the same stock. We assume no transaction cost for simplicity.

**C.3. Results**

Figure 2 shows the cumulative performance of these portfolios in terms of the net value (starting at 1).

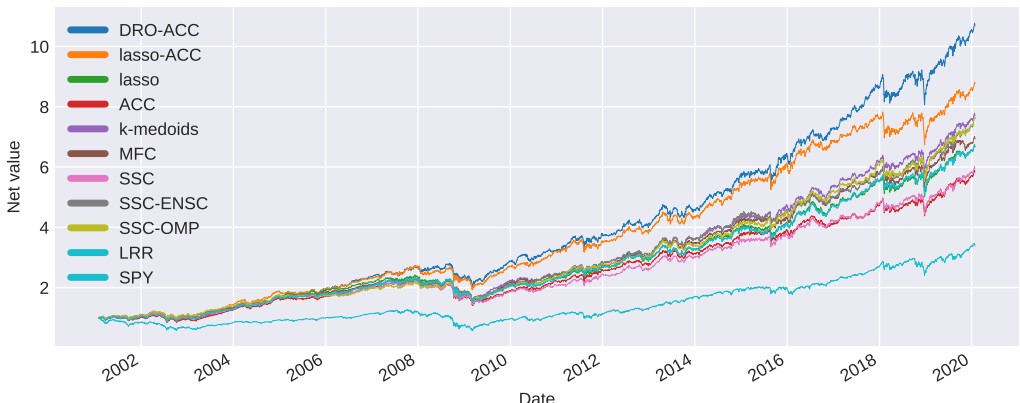

*Figure 2.* Cumulative performance of portfolios constructed by different methods.

Figure 3 shows the clustering results obtained by the DRO-ACC clustering method on Feb 1st, 2019, date of the last portfolio update in our experiment. The clusters are first ordered by the 6 major clusters from DRO, and then by size within each major cluster. In other words, Clusters 1 to 6 are from the first DRO major cluster, Clusters 7 to 12 are from the second, and so on.

One observation that immediately stands out is the similarity to GICS sectors in the 2nd through 6th DRO major clusters (noting the colors starting from Cluster 7). Each of the 2nd through 5th major clusters covers a different sector and often includes most companies in that sector. The last DRO cluster (Clusters 31-36) includes two sectors, namely, Consumer Discretionary and Consumer Staples, which are closely related to each other. In comparison, clusters within the first DRO cluster tend to be larger, especially Cluster 1. They also include companies from many different sectors, such as Communication Services, Consumers Discretionary, Industrials, and Information Technology. Intuitively, these sectors appear to be more closely related to the notion of the "day-to-day" economy, than some sectors represented by the other major clusters, like Real Estate, Energy, Utilities, and Financials. The reason why the sectors in the DRO clusters 2 through 6 (Clusters 7 through 36 in Figure 3) stand out is likely because they are the most distinguishable sectors from the rest of the

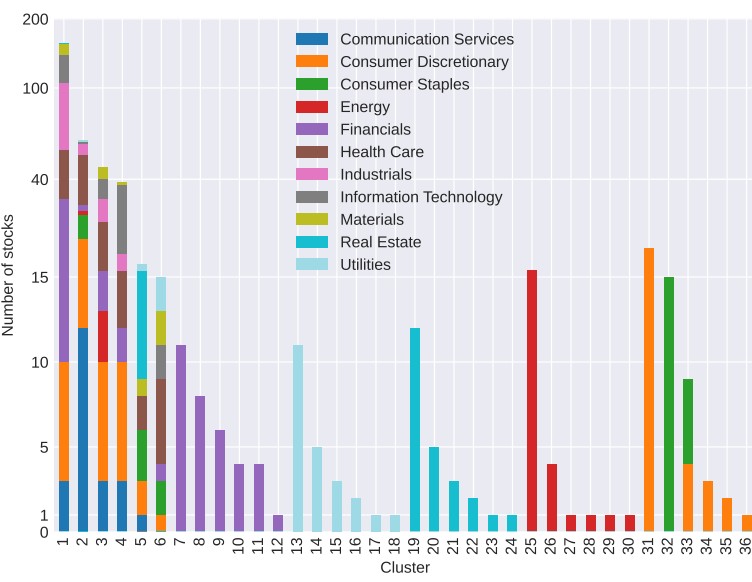

*Figure 3.* DRO-ACC clustering results on 2019-02-01 compared with GICS sectors

market. DRO being able to single them out in the first stage of clustering guarantees that a sufficient number of stocks are selected from each of these distinguishable sectors, which may facilitate diversification and lead to the good performance of the DRO-ACC portfolio.

### C.4. Additional portfolio backtesting results

According to Table 3, the DRO-ACC portfolio outperforms the others significantly in many important metrics, including Sharpe, Sortino, and Calmar ratios, annualized return, maximum drawdown, and recovery time, while it performs similarly to the best performers in other metrics.

*Table 3.* Performance Metrics of the Minimum Variance Portfolios; DRO-ACC Creates 6x6 Clusters, Other Algorithms Create 36 Clusters

| | DRO-ACC | lasso-ACC | ACC | $k$-medoids | SSC | SSC-ENSC | SSC-OMP | MFC | LRR | SPY |
|---|---|---|---|---|---|---|---|---|---|---|
| **Ending VAMI** | **10694.3** | 8757.96 | 5830.69 | 7701.07 | 5958.39 | 7617.9 | 7622.97 | 6935.13 | 6680.2 | 3376.29 |
| **Max Drawdown** | **27.72%** | 29.8% | 36.24% | 31.87% | 34.91% | 32.48% | 31.6% | 34.22% | 36.02% | 55.19% |
| **Peak-To-Valley** | 2008-09-08 - 2009-03-09 | 2007-12-13 - 2009-03-09 | 2007-12-10 - 2009-03-09 | 2007-12-10 - 2009-03-09 | 2007-12-10 - 2009-03-09 | 2007-12-10 - 2009-03-09 | 2007-12-10 - 2009-03-11 | 2007-12-10 - 2009-03-09 | 2007-12-10 - 2009-03-09 | 2007-10-09 - 2009-03-09 |
| **Recovery** | **194** Days | 216 Days | 446 Days | 250 Days | 540 Days | 379 Days | 384 Days | 262 Days | 472 Days | 869 Days |
| **Sharpe Ratio** | **1.05** | 0.96 | 0.77 | 0.9 | 0.79 | 0.89 | 0.89 | 0.85 | 0.8 | 0.36 |
| **Sortino Ratio** | **1.73** | 1.57 | 1.25 | 1.46 | 1.28 | 1.45 | 1.45 | 1.38 | 1.3 | 0.56 |
| **Calmar Ratio** | **0.48** | 0.41 | 0.27 | 0.36 | 0.28 | 0.35 | 0.36 | 0.31 | 0.29 | 0.12 |
| **Ann. Volatility** | 12.73% | 12.62% | 12.59% | 12.69% | **12.55%** | 12.69% | 12.67% | 12.6% | 13.2% | 18.63% |
| **Ann. Downside Volatility** | 7.71% | 7.72% | 7.77% | 7.81% | **7.70%** | 7.79% | 7.8% | 7.78% | 8.13% | 11.81% |
| **Correlation** | 0.78 | **0.77** | 0.8 | 0.8 | 0.8 | 0.78 | 0.78 | 0.8 | 0.78 | 1.0 |
| **Beta** | 0.53 | **0.52** | 0.54 | 0.55 | 0.54 | 0.53 | 0.53 | 0.54 | 0.55 | 1.0 |
| **Annualized Return** | **13.32%** | 12.14% | 9.75% | 11.38% | 9.88% | 11.31% | 11.32% | 10.76% | 10.54% | 6.63% |
| **Positive Periods** | 2608 (54.63%) | 2608 (54.63%) | 2611 (54.69%) | 2595 (54.36%) | 2595 (54.36%) | 2607 (54.61%) | 2617 (54.82%) | 2595 (54.36%) | 2597 (54.40%) | **2626 (55.01%)** |
| **Negative Periods** | 2166 (45.37%) | 2166 (45.37%) | 2163 (45.31%) | 2179 (45.64%) | 2179 (45.64%) | 2167 (45.39%) | 2157 (45.18%) | 2179 (45.64%) | 2177 (45.60%) | **2148 (44.99%)** |

Below we present Sharpe ratios of the portfolios in backtesting with different values for $K_1 = K_2$. As shown in Table 4, with annual portfolio updates, the DRO-ACC performs well with $3 \times 3$ and $4 \times 4$ clusters. With $5 \times 5$ clusters, the DRO-ACC portfolio underperforms the other portfolios in Sharpe ratio but still outperforms the benchmark SPY.

*Table 4.* Sharpe Ratio of portfolios with different numbers of clusters, updated annually.

| # clusters | $3 \times 3 = 9$ | $4 \times 4 = 16$ | $5 \times 5 = 25$ | $6 \times 6 = 36$ |
|---|---|---|---|---|
| **DRO-ACC** | 0.87 | 0.87 | 0.73 | **1.05** |
| **lasso-ACC** | 0.89 | **0.9** | 0.86 | 0.96 |
| **ACC** | 0.9 | 0.85 | 0.79 | 0.77 |
| **k-medoids** | 0.82 | 0.85 | 0.84 | 0.9 |
| **MFC** | 0.85 | 0.85 | 0.85 | 0.85 |
| **SSC** | 0.83 | 0.78 | 0.81 | 0.79 |
| **SSC-ENSC** | 0.92 | 0.88 | 0.84 | 0.89 |
| **SSC-OMP** | **0.94** | 0.81 | **0.93** | 0.89 |
| **LRR** | 0.8 | 0.8 | 0.8 | 0.8 |
| **SPY** | 0.36 | 0.36 | 0.36 | 0.36 |

We also present the Sharpe ratios with monthly and quarterly portfolio updates. This means that the stocks are selected monthly/quarterly and weights re-calculated using the newest clustering results. As shown in Tables 5 and 6, DRO-ACC portfolios are also robust to the stock selection and allocation update frequency, as they consistently outperform the benchmark and tend to achieve close to the best performance with both monthly and quarterly updates.

*Table 5.* Sharpe Ratio of portfolios with different numbers of clusters, updated monthly.

| # clusters | $3 \times 3 = 9$ | $4 \times 4 = 16$ | $5 \times 5 = 25$ | $6 \times 6 = 36$ |
|---|---|---|---|---|
| **DRO-ACC** | 0.8 | 0.83 | 0.88 | 0.89 |
| **lasso-ACC** | 0.73 | **0.98** | 0.69 | 0.77 |
| **ACC** | 0.65 | 0.84 | 0.77 | 0.74 |
| **k-medoids** | 0.78 | 0.8 | 0.81 | 0.87 |
| **MFC** | 0.75 | 0.75 | 0.75 | 0.75 |
| **SSC** | 0.79 | 0.82 | 0.74 | 0.84 |
| **SSC-ENSC** | **0.94** | 0.83 | **0.91** | 0.84 |
| **SSC-OMP** | 0.82 | 0.89 | 0.87 | **0.91** |
| **LRR** | 0.79 | 0.79 | 0.79 | 0.79 |
| **SPY** | 0.36 | 0.36 | 0.36 | 0.36 |

## D. Additional Numerical Results

### D.1. Implementation details

We implemented DRO, Lasso, ACC, $k$-medoids, and MFC by ourselves. We used open-source code to run the other methods, with default hyperparameter settings. Specifically, we obtained the code for SCC from `https://github.com/abhinav4192/sparse-subspace-clustering-python`; SSC-OMP and EnSC from `https://github.com/ChongYou/subspace-clustering`; LRR from `https://github.com/barbosaaob/lrr`; co-clustering from the Python library `Coclust` (Role et al., 2019).

### D.2. Simulation analysis

We still create a total of $K = 25$ clusters among $d = 500$ variables, and generate $n = 250$ i.i.d. samples for each experiment, fixing $\beta_H(i) = 0$ for all $i$, which means no hidden factor. The number of factors controlling each cluster $k$ is randomly chosen from 1 to $m_k - 1$, where $m_k$ is the number of variables in cluster $k$. As a demonstration, we temporarily keep the

*Table 6.* Sharpe Ratio of portfolios with different numbers of clusters, updated quarterly.

| # clusters | $3 \times 3 = 9$ | $4 \times 4 = 16$ | $5 \times 5 = 25$ | $6 \times 6 = 36$ |
|---|---|---|---|---|
| **DRO-ACC** | **0.91** | 0.83 | 0.78 | 0.87 |
| **lasso-ACC** | 0.85 | **0.93** | 0.8 | 0.85 |
| **ACC** | 0.76 | 0.82 | 0.88 | 0.84 |
| **k-medoids** | 0.72 | 0.78 | 0.78 | 0.82 |
| **MFC** | 0.79 | 0.79 | 0.79 | 0.79 |
| **SSC** | 0.66 | 0.8 | 0.77 | 0.75 |
| **SSC-ENSC** | 0.79 | 0.81 | 0.79 | 0.87 |
| **SSC-OMP** | **0.91** | 0.87 | **0.96** | **0.89** |
| **LRR** | 0.8 | 0.8 | 0.8 | 0.8 |
| **SPY** | 0.36 | 0.36 | 0.36 | 0.36 |

noise level fixed for all variables at $\mathrm{Var}(U_i) = 0.1$ and Figure 4a shows the true clustering structure, and Figure 4b shows a heatmap of the sample correlation matrix. The blocks along the diagonal of Figure 4b are nearly indistinguishable. Figures 5a and 5b show the $C$ matrices from DRO and Lasso, respectively. Similar to the previous experiments, both methods can extract the blocks by making them visually more prominent, with Lasso extracting a sparse $C$ matrix while DRO keeps more entries in the matrix but at lower magnitudes.

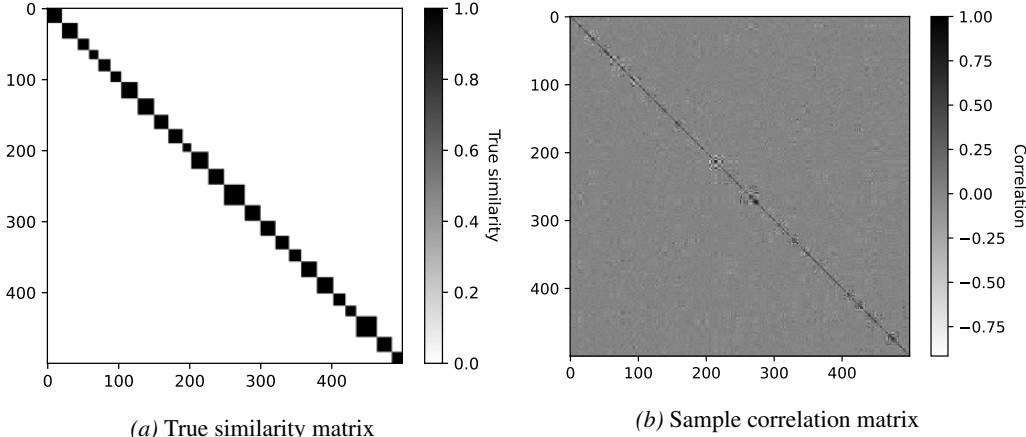

*(a)* True similarity matrix        *(b)* Sample correlation matrix

*Figure 4.* The heatmap of the true similarity matrix (a) and the sample correlation matrix (b), with $\beta_H = 0$, $\sigma_e^2 = 0.1$, and $d_k$ randomly chosen. Variables in the same cluster have similarity 1 and otherwise 0.

Table 7 shows the average AMI of each clustering method over the 10 random trials. DRO performs the best and much better compared to Lasso. Both methods still outperform ACC and $k$-medoids. As expected, neither ACC nor $k$-medoids can meaningfully recover the clusters in this experiment due to the additional complexity in the underlying model.

We now test different noise levels by setting $\sigma_e^2 = 0.1, 0.2, \ldots, 2.0$ and repeating the experiment on 10 random trials for each value of $\sigma_e^2$. These values of $\sigma_e^2$ represent signal-to-noise ratios from 10:1 to 1:2. The average AMI of each method is shown in Figure 6. Overall, the average AMI decays for all methods as the level of noise increases. The DRO subspace clustering methods perform similarly with Lasso, and both consistently outperform ACC and $k$-medoids in this experiment.

### D.3. Sensitivity analysis

We present full results of the three ablation studies described in Section 4. All experiments use the same data-generating process ($d = 500$, $n = 250$, $K = 25$, $\beta_H^2(i) \sim U[0, 0.5]$, $\mathrm{Var}(U_i) \sim U[0, 0.5]$) over 10 random trials with seeds 2021–2030.

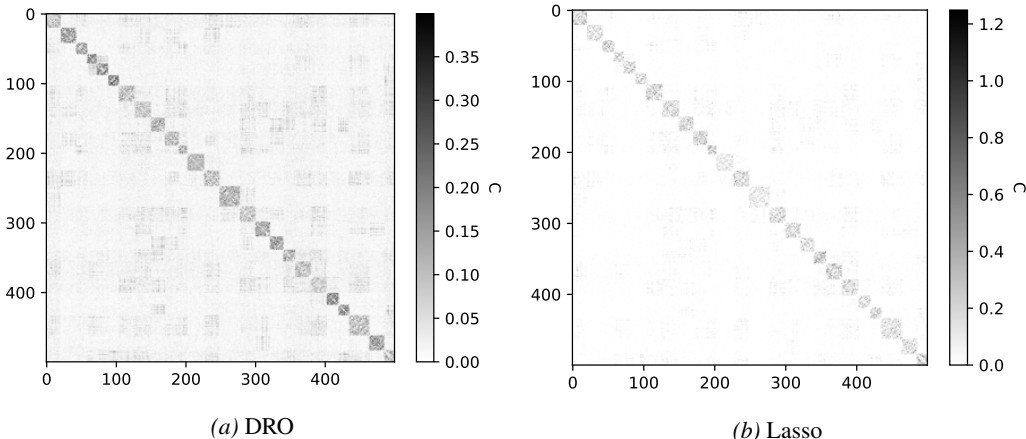

*(a)* DRO            *(b)* Lasso

*Figure 5.* The heatmap of the $C$ matrices for DRO (a) and Lasso (b), with $\beta_H = 0$, $\sigma_e^2 = 0.1$, and $d_k$ randomly chosen.

*Table 7.* Average AMI of different clustering methods compared with ground truth, over 10 different random trials.

| Method | Average AMI |
|---|---|
| DRO | **0.96** |
| Lasso | 0.94 |
| ACC | 0.34 |
| $k$-medoids | 0.55 |
| MFC | 0.57 |
| SSC | 0.82 |
| SSC-ENSC | 0.92 |
| SSC-OMP | 0.27 |
| LRR | 0.21 |
| Co-Clustering | 0.05 |

**ADMM penalty parameter $\rho$.** In the ADMM algorithm (Equation (9)–Equation (10)), the penalty parameter $\rho$ is initialized and then adaptively adjusted during optimization. Table 8 shows that the clustering performance is stable across initial $\rho$ values spanning three orders of magnitude, demonstrating that the adaptive $\rho$-update scheme is effective.

*Table 8.* Sensitivity to the initial ADMM penalty parameter $\rho$. Reported: mean $\pm$ std of AMI over 10 trials.

| $\rho$ | AMI |
|---|---|
| 0.01 | $0.912 \pm 0.021$ |
| 0.10 | $0.910 \pm 0.028$ |
| 0.50 | $0.920 \pm 0.025$ |
| 1.00 | $0.920 \pm 0.025$ |
| 2.00 | $0.917 \pm 0.022$ |
| 5.00 | $0.920 \pm 0.025$ |
| 10.00 | $0.919 \pm 0.024$ |

**Misspecified number of clusters $K$.** In practice, the true number of clusters is often unknown. Table 9 reports the clustering performance when spectral clustering is applied with a misspecified $K$, while the DRO regression coefficients $B$ are computed under the default settings. The method is robust to slight overestimation of $K$ (e.g., $K = 27$ yields AMI $= 0.919$, comparable to the true $K = 25$), while underestimation leads to a more rapid degradation.

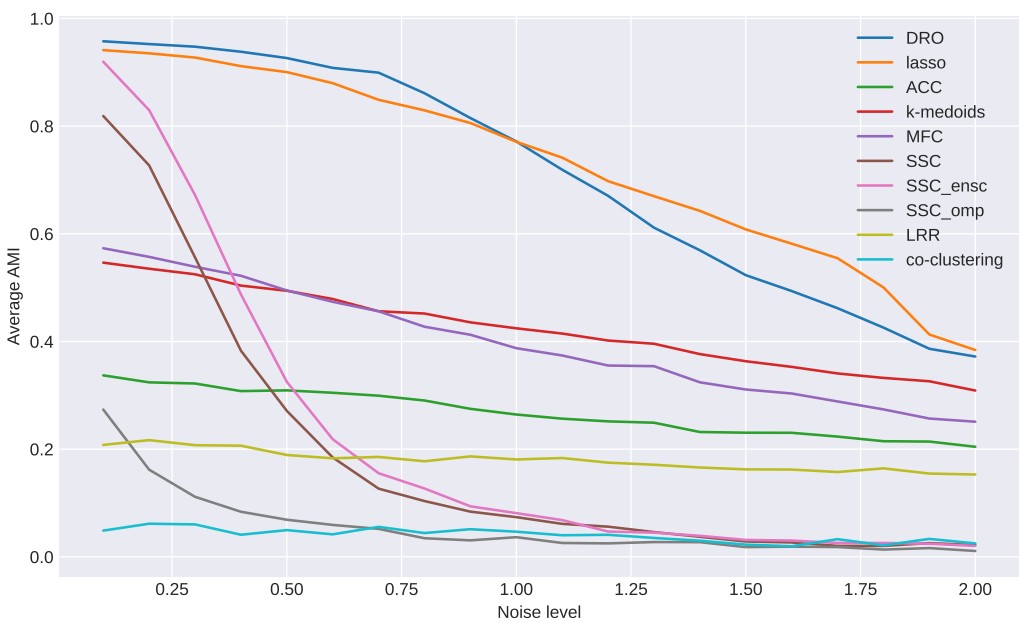

*Figure 6.* Comparison of average AMI between different clustering methods, with $\beta_H = 0$ and $d_k$ randomly chosen.

*Table 9.* Sensitivity to misspecified number of clusters $K$ (true $K = 25$). Reported: mean $\pm$ std of AMI over 10 trials.

| $K$ | AMI |
|---|---|
| 10 | $0.463 \pm 0.026$ |
| 15 | $0.682 \pm 0.032$ |
| 20 | $0.826 \pm 0.021$ |
| 23 | $0.884 \pm 0.023$ |
| **25** | $\mathbf{0.920 \pm 0.025}$ |
| 27 | $0.919 \pm 0.020$ |
| 30 | $0.903 \pm 0.017$ |
| 35 | $0.857 \pm 0.015$ |
| 40 | $0.814 \pm 0.015$ |

**Confidence level** $1 - \alpha$**.** The parameter $\alpha$ controls the confidence level used to calibrate the DRO uncertainty radius $\delta$ (Appendix B). Table 10 shows that the clustering performance is virtually unchanged across a wide range of $\alpha$ values, confirming that the DRO method is insensitive to this tuning parameter.

*Table 10.* Sensitivity to the confidence level parameter $\alpha$. Reported: mean $\pm$ std of AMI over 10 trials.

| $\alpha$ | Confidence level | AMI |
|---|---|---|
| 0.001 | 99.9% | $0.916 \pm 0.024$ |
| 0.01 | 99% | $0.916 \pm 0.030$ |
| **0.05** | **95%** | $\mathbf{0.920 \pm 0.025}$ |
| 0.10 | 90% | $0.921 \pm 0.026$ |
| 0.20 | 80% | $0.923 \pm 0.029$ |

## Additional Figures and Tables

This section provides additional figures and tables that complement the main text.

### D.4. Toy multi-factor block model example

We start with a toy multi-factor block model instance (five variables, three factors, and two clusters) to make the induced near-block covariance structure concrete.

*Example* 1. Let $X = (X_1, X_2, X_3, X_4, X_5)$ be a random vector in $\mathbb{R}^5$. Consider a partition $G := \{G_1, G_2\} := \{\{1, 2, 3\}, \{4, 5\}\}$, where the first three random variables are in the same cluster, and the last two in the same cluster. Let $d_1 = 2$, $d_2 = 1$, i.e. the first cluster is controlled by two factors, and the second cluster by one factor. Denote the latent factors by $F_G$, and $F_G = (F_G^{1\top}, F_G^{2\top})^\top = (F_1^\top, F_2^\top, F_3^\top)^\top$, where $F_1$, $F_2$, $F_3$ represent the three latent factors, whose covariance matrix is

$$\Sigma_F = \begin{bmatrix} 1 & 0.1 & 0.5 \\ 0.1 & 1 & 0.5 \\ 0.5 & 0.5 & 1 \end{bmatrix}.$$

Each random variable $X_i$ only loads on the latent factors controlling the corresponding cluster. The loading matrix $A \in \mathbb{R}^{5\times3}$ is

$$A = \begin{bmatrix} 0.4 & 0.6 & 0 \\ 0.7 & 0.3 & 0 \\ 0.4 & 0.6 & 0 \\ 0 & 0 & 0.8 \\ 0 & 0 & 0.7 \end{bmatrix}.$$

Let the covariance of the idiosyncratic components, denoted by $\Gamma$, be a diagonal matrix with diagonal entries $(0.1, 0.1, 0.1, 0.1, 0.1)$, then the covariance of the random vector $X$ can be calculated:

$$\Sigma = A\Sigma_F A^\top + \Gamma = \begin{bmatrix} 0.722 & 0.478 & 0.586 & 0.4 & 0.35 \\ 0.478 & 0.722 & 0.514 & 0.4 & 0.35 \\ 0.586 & 0.514 & 0.668 & 0.4 & 0.35 \\ 0.4 & 0.4 & 0.4 & 0.74 & 0.56 \\ 0.35 & 0.35 & 0.35 & 0.56 & 0.59 \end{bmatrix}.$$

We observe that the covariance matrix $\Sigma$ displays a near-block structure. Figure 7b illustrates this observation with a heatmap. One can see four blocks and similar values within the blocks. The $3 \times 3$ block on the top left and the $2 \times 2$ block on the bottom right have slightly higher values than the off-diagonal blocks.

Using the same toy example, we can compute in closed form the population-level nodewise regression coefficients $B$, which correspond to the optimizer of (3) in the limit as $n$ approaches infinity. The symmetrized similarity matrix $C$ calculated from the optimal $B$, i.e., $C := B_{abs}^\top + B_{abs}$ is shown below.

$$C = \begin{bmatrix} 0 & 0.141 & 1.357 & 0.107 & 0.085 \\ 0.141 & 0 & 0.865 & 0.179 & 0.145 \\ 1.357 & 0.865 & 0 & 0.101 & 0.079 \\ 0.107 & 0.179 & 0.101 & 0 & 1.530 \\ 0.085 & 0.145 & 0.079 & 1.530 & 0 \end{bmatrix}.$$

Figure 7 visualizes and compares $C$ and $\Sigma$ in heatmaps. We can see that in $C$, the two blocks along the diagonal have much larger values than the off-diagonal blocks, whereas, in $\Sigma$, the same blocks are more difficult to distinguish.

### D.5. Visualization for simulation experiments

To provide a qualitative view of the clustering structure, we visualize the true similarity matrix, the sample correlation matrix, and the similarity matrices $C$ extracted by the subspace clustering methods. Figure 8a shows the true clustering structure among the $d = 500$ variables, and Figure 8b shows a heatmap of the sample correlation matrix. The blocks along the diagonal of Figure 8b are very difficult to distinguish, likely due to the presence of multiple group-specific factors. Figures 9a and 9b show the $C$ matrices from DRO and Lasso, respectively. We observe that the blocks along the diagonal are more prominent visually with both methods. DRO maintains more connections than Lasso in the $C$ matrix, reflected in

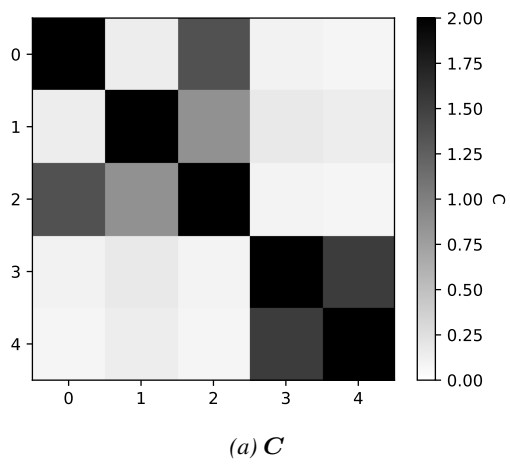

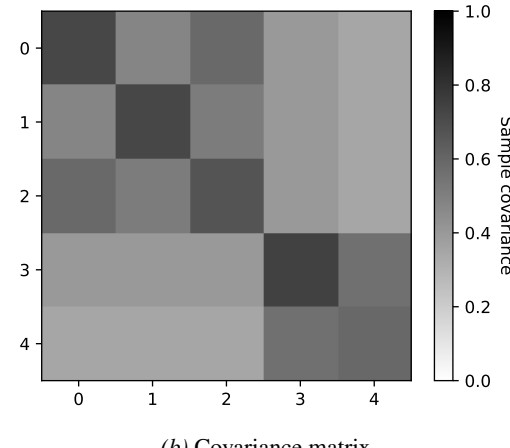

*(a) $C$*

*(b) Covariance matrix*

*Figure 7.* The heatmap of $C$ compared with that of the covariance matrix from population-level nodewise regression on variables in Example 1. Diagonals of $C$ are filled with value 2 to facilitate the visualization.

the light grey background off the diagonals. In contrast, Lasso leaves more blanks on the off-diagonals. On the diagonals, the blocks are also darker in DRO than Lasso. This is consistent with our intuition that DRO does not artificially pursue sparsity and thus has the advantage of keeping more true connections while weakening, instead of eliminating, irrelevant connections.

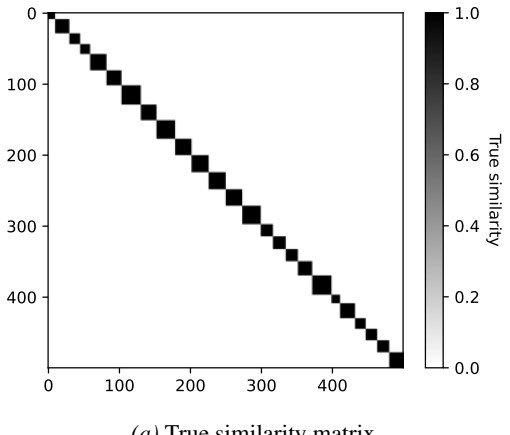

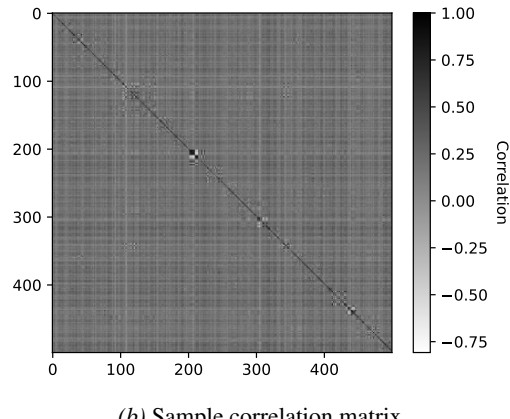

*(a) True similarity matrix*

*(b) Sample correlation matrix*

*Figure 8.* The heatmap of the true similarity matrix (a) and the sample correlation matrix (b), with $\beta_H(i)^2$ and $\mathrm{Var}(\varepsilon_i)$ drawn independently and uniformly from $[0, 0.5]$, and $d_k$ randomly chosen. Variables in the same cluster have similarity 1 and otherwise 0.

### D.6. Wall-clock runtime comparison

Table 11 reports a representative wall-clock runtime comparison (in seconds) for estimating $K = 25$ clusters among $d = 500$ variables from $n = 250$ observations.

*Table 11.* Wall-clock runtime comparison between clustering algorithms

| DRO - ADMM | DRO - CVX | Lasso | ACC | $k$-medoids | MFC |
|---|---|---|---|---|---|
| 52.5 | 331.5 | 1916.6 | 4.1 | 1.8 | 66.2 |

| SSC | SSC-ENSC | SSC-OMP | LRR | Co-Clustering |
|---|---|---|---|---|
| 691.9 | 355.6 | 1521.4 | 42.7 | 0.3 |

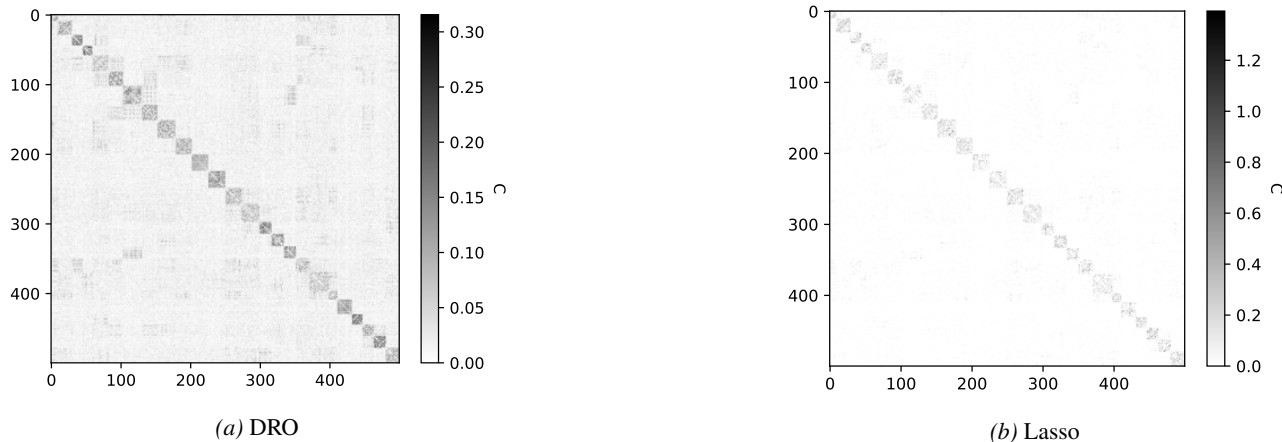

*(a)* DRO

*(b)* Lasso

*Figure 9.* The heatmap of the $C$ matrices for DRO (a) and Lasso (b), with $\beta_H(i)^2$ and $\mathrm{Var}(U_i)$ drawn independently and uniformly from $[0, 0.5]$, and $d_k$ randomly chosen.

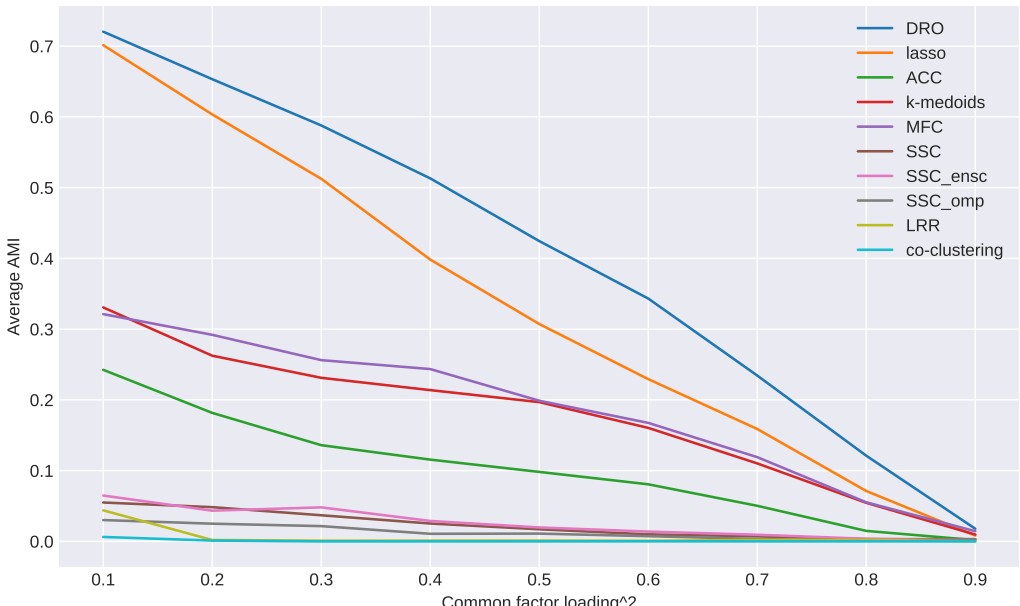

*Figure 10.* Comparison of average AMI between different common factor loadings, with $\mathrm{Var}(U_i) = 1.0$ and $d_k$ randomly chosen.

### D.7. Face clustering results on three standard splits

For completeness, Table 12 reports the AMI results on the three standard 10-subject splits used in Section 5.

*Table 12.* AMI of different algorithms on the Extended Yale B dataset

| Metric | DRO | Lasso | $k$-medoids | MFC | ACC |
|---|---|---|---|---|---|
| Mean | **0.576** | 0.410 | 0.099 | 0.141 | 0.001 |
| Median | **0.565** | 0.422 | 0.092 | 0.140 | 0.001 |

| Metric | SSC | SSC-ENSC | SSC-OMP | LRR | Co-Clustering |
|---|---|---|---|---|---|
| Mean | 0.095 | 0.230 | 0.014 | -0.018 | 0.003 |
| Median | 0.087 | 0.221 | 0.014 | -0.022 | 0.004 |

