# OpenReview forum: "Variable Clustering via Distributionally Robust Nodewise Regression"
_ICML.cc/2026/Conference — ICML 2026 regular_

### Official Review · Reviewer_DvfT · 2026-03-06

**Soundness:** 3
**Presentation:** 3
**Significance:** 2
**Originality:** 2
**Overall Recommendation:** 3
**Confidence:** 4

**Summary:**

This paper studies the variable clustering problem via a multi-factor block model. The paper proposes a distributionally robust optimization (DRO) formulation for the nodewise regression method. The convex relaxed version of DRO nodewise regression is then solved by an ADMM algorithm. Several numeric examples are shown to demonstrate the empirical benefit of the proposed methods over Lasso and other subspace clustering methods.

**Compliance With Llm Reviewing Policy:**

Affirmed.

**Final Justification:**

This paper presented a convex relaxation of the DRO nodewise regression for clustering variables. A worst-case relaxation bound was derived. The rebuttal partially addressed some of my concerns, but it remains non-trivial to study the finite-sample average-case tightness and guarantees of such a convex formulation. I raised my score from 2 to 3, reflecting my view that there should be substantial room for this paper to improve, and my opinion of this paper is borderline.

**Key Questions For Authors:**

- How tight is the convex relaxation in Theorem 3.1 of the DRO formulation (7)?
- A local convergence analysis would be helpful to justify the convergence of the proposed ADMM algorithm.
- It would be helpful to give a sample complexity bound for estimating the coefficient matrix $B$. Even though this does not directly translate to the clustering accuracy, it provides an intermediate measure of statistical errors for the proposed approach.
- Is the K-means clustering algorithm on page 4 correct? It seems that the assignment operation $z_1, \dots, z_d$ should be inside $\sum_{i=1}^d$.

**Limitations:**

yes

**Strengths And Weaknesses:**

Strength

- The paper is well written and easy to follow.
- Combining the DRO with nodewise regression to identify group structure in variables to enhance generalization is intuitive.
- ADMM algorithm is computationally cheap than directly solving the convex relaxed formulation of variable clustering.

Weakness
- I don’t see how the DRO formulation equation (7) can ensure the low-rank structure of the coefficient matrix $B$, which is used in the subsequent spectral clustering algorithm to identiy the variable clusters.
- It is unclear the tightness of the convex relaxation in Theorem 3.1 of the DRO formulation (7). Theorem 3.1 is an algebraic inequality in the worst-case. The primary setup of the paper is i.i.d. data sampling from a continuous distribution (c.f. line 162-164, page 3). An average-case bound under standard noise model for the data generation mechanism can be helpful to justify the relaxation tightness.
- The ADMM breaks the convexity. There is a lack of convergence analysis for the proposed algorithm, and it is unclear how the initialization and the penalty parameter impact the convergence.
- Choice of the radius parameter $\delta$ still depends on the confidence level, which in certain sense is a tuning parameter and should have a trade-off effect on the accuracy and out-of-sample performance.
- The paper focuses on the estimation of the coefficient matrix $B$, while mentioned that spectral clustering is used to recover the variable cluster structure. The second step of spectral clustering is commonly seen as a rounding procedure to recover the low-rank structure. It is unclear how this step impact the performance if one substitute with other clustering methods based on the estimated similarity matrix.

---

> ### Author Rebuttal · Authors · 2026-03-29
>
> We thank the reviewer for the careful reading and constructive comments. Below we address each concern.
>
> ### 1. On whether the DRO formulation enforces low-rank structure of $\mathbf{B}$
>
> Our method does not aim to estimate a low-rank coefficient matrix $\mathbf{B}$. As in standard self-expressive subspace clustering methods, such as SSC (Elhamifar and Vidal, 2013), the nodewise regression step returns a coefficient matrix $\mathbf{B}$ that serves as a similarity matrix for clustering. The spectral norm penalty in our convex relaxation does not promote low-rank structure. It arises from the DRO formulation to stabilize nodewise regression under uncertainty. We will revise the exposition to make it clearer.
>
> ### 2. On the tightness of the convex relaxation in Theorem 3.1
>
> Thank you for this important question. We have strengthened the theorem and shown that the relaxation is tight up to a factor of 2:
> $f(\mathbf{B})/2
> \le
> \sup_{\mathbb{P}: D_c(\mathbb{P}, \mathbb{P}\_n)
> \le \delta}
> \mathbb{E}_{\mathbb{P}}\\|X-\mathbf{B}^\top X\\|_2^2
> \le
> f(\mathbf{B})$, where $f(\mathbf{B})=
> \left(
> \frac{1}{\sqrt{n}}\\|\mathbf{X}-\mathbf{X}\mathbf{B}\\|_F
> +
> \sqrt{\delta}\\|\mathbf{I}-\mathbf{B}\\|_2
> \right)^2$. We will add this to the revision.
>
> ### 3. On ADMM, convexity, convergence, initialization, and the penalty parameter
>
> We respectfully clarify that the ADMM procedure does not break convexity. The optimization problem we solve is convex, and ADMM is used only as a standard decomposition method for a convex two-block problem. This is the setting in the classical ADMM literature (Boyd et al., 2010). There is also a substantial convergence literature for ADMM in convex settings, including global convergence guarantees (He and Yuan, 2012).
>
> We adopt zero initialization for $\mathbf{B}_2^0$ and $\mathbf{\Lambda}^0$, together with the varying penalty scheme of Boyd et al. (2010), which adaptively adjusts $\rho$ during optimization. In the simulation experiments, we added an ablation on the sensitivity to the initial value of $\rho$ over three orders of magnitude, from $0.01$ to $10$. The clustering performance remains essentially unchanged. For numerical results, please kindly refer to the table in our response to Reviewer f2Ca.
>
>
>
> ### 4. On the radius parameter $\delta$
>
> We do not tune $\delta$ directly. Appendix B gives a principled rule that selects $\delta$ via a confidence level $1-\alpha$, which is interpretable, unit-free, and less instance-dependent than the raw penalty $\delta$. See the first paragraph of Appendix B for more discussions on this. Our experiments used the standard choice $\alpha = 0.05$. We also added an ablation over $\alpha$ and found very similar performance throughout:
>
> |$\alpha$|AMI (mean $\pm$ std)|
> |-|-|
> |0.001|$0.916 \pm 0.024$|
> |0.01|$0.916 \pm 0.030$|
> |0.05|$0.920 \pm 0.025$|
> |0.10|$0.921 \pm 0.026$|
> |0.20|$0.923 \pm 0.029$|
>
>
>
> ### 5. On the role of spectral clustering and possible alternatives
>
> This is a very good point. Our paper focuses primarily on the first step estimating a similarity matrix through DRO nodewise regression. The second step then uses spectral clustering, following standard practice in subspace clustering (Elhamifar and Vidal, 2013). This is not the only possible choice; other clustering methods on the learned similarity matrix could also be used. We chose spectral clustering to follow the convention and make comparisons cleaner. We will clarify this in the paper.
>
> ### 6. On sample complexity for estimating $\mathbf{B}$
>
> We agree that a finite-sample estimation bound for $\mathbf{B}$ would be valuable. The current paper provides a worst-case DRO interpretation, a convex relaxation with a factor-2 guarantee, and empirical clustering results, but not yet a non-asymptotic error bound for $\mathbf{B}$. We will clarify this limitation and view it as an important direction for future work.
>
> ### 7. On the $k$-means expression on page 4
>
> We thank the reviewer for the question. We believe the current formulation is already correct:
> $\min_{z\_1,...,z\_d\in[K],\~\\mu\_1,...,\\mu\_K\in\mathbb{R}\^n}
> \\{
> \sum\_{i=1}\^d \\|x\_i-\\mu_{z\_i}\\|\_2\^2
> \\}$. The joint minimization over assignments and centroids is equivalent to the nested formulation that puts assignments inside the summation:
> $
> \min_{\\mu\_1,...,\\mu\_K\in\mathbb{R}\^n}
> \\{
> \sum\_{i=1}\^d \min_{z\_i\in[K]} \\|x\_i-\\mu_{z\_i}\\|\_2\^2
> \\}$.
> We are happy to adopt the second form to avoid possible ambiguity.
>
> ### References
> Boyd et al., 2010. Distributed optimization and statistical learning via the alternating direction method of multipliers. Foundations and Trends in Machine Learning.
>
> Elhamifar and Vidal, 2013. Sparse subspace clustering: Algorithm, theory, and applications. IEEE transactions on pattern analysis and machine intelligence.
>
> He and Yuan, 2012. On the O(1/n) convergence rate of the Douglas–Rachford alternating direction method. SIAM Journal on Numerical Analysis.

---

> > ### Author Rebuttal · Reviewer_DvfT · 2026-04-02
> >
> > Thanks for your rebuttal. Some of my concerns have been clarified. However, I feel that there is still room to improve the constant-accuracy relaxation bound and the lack of average-case statistical guarantees. I will raise my score slightly.

---

### Official Review · Reviewer_T997 · 2026-03-08

**Soundness:** 3
**Presentation:** 1
**Significance:** 3
**Originality:** 3
**Overall Recommendation:** 4
**Confidence:** 1

**Summary:**

The authors connect multi-factor block model to a subspace clustering problem, and propose a solution based on nodewise regression

**Compliance With Llm Reviewing Policy:**

Affirmed.

**Final Justification:**

I am happy with the authors' response and will raise my score.

**Key Questions For Authors:**

- Can you say a bit more about how you implement spectral clustering? There are a lot of choices there. I could not find the code (sorry if I missed it)
- is there any intuition to be had about why |b_ij| is the natural measure of similarity, rather than e.g. b^2_ij? I ask because, as far as I know, the spectral clustering results would not be the same.
- Related to the above, since the clustering is entirely "outsourced" to the spectral clustering algorithm, is it better to understand the "real" contribution of this paper as a better measure of variable similarity? If that is the case, would it not be more direct to compare the quality of similarity matrices obtained by different methods rather than the clusterings?

**Limitations:**

yes

**Strengths And Weaknesses:**

The contribution tackles an important problem, variable clustering, with seemingly strong computational methodology.

The real data aspect is extremely limited, which is a shame given the fairly broad applicability of this sort of method. I think it would have been nice to see some variable clusters and their meanings in the main text. (I know there is a little in the appendix.)

I did not find the paper very easy to follow; perhaps this is a little subjective because I somehow struggle to think of clustering dimensions as easily as clustering observations. I do think the paper could help the reader a bit more, e.g. a small example giving values for n, d, D, etc and neater notation. For example, why do we have to carry the G subscript around on F_G?

I have the impression this is generally good quality research, but it is definitely lacking on presentation/explanation and real data experiments.

---

> ### Author Rebuttal · Authors · 2026-03-29
>
> We thank the reviewer for the thoughtful comments and for recognizing the importance of the variable clustering problem and the strength of the computational methodology. Below we address each point.
>
> **1. On the real-data evaluation.**
> We would like to clarify that the initial submission already included one real-data experiment in the main text (face clustering, Section 5), where our DRO method was compared against a broad set of benchmarks from the literature, including $k$-medoids, MFC, ACC, SSC, SSC-EnSC, SSC-OMP, LRR, and Co-Clustering. In addition, Appendix C included a second real-data experiment on financial stock clustering, which was designed to illustrate the interpretability of the recovered clusters. Our clusters are closely aligned with the sectors defined by Global Industry Classification Standard (GICS), see Appendix C.4.
>
> We agree, however, that the real-data discussion in the main body should be strengthened. Due to page limits, we placed the financial-data experiment in the appendix in the initial submission. Since the final version allows one additional page, we will move the main results of the financial-data experiment into the main text so that both empirical performance and interpretability are more visible.
>
> **2. On presentation, notation, and accessibility.**
> We appreciate this feedback and agree that the paper can do more to help readers. In the revision, we will improve the exposition in three ways:
>
> - add a small running example clarifying the roles of $n$, $d$, etc.: in financial stock clustering from daily returns, $n$ is the number of trading days in the data, $d$ is the number of stocks, $K$ is the number of sectors, the $k$-th sector is a group of stocks driven by $d_k$ common factors, $D = d_1 + \cdots + d_K$ is the total number of factors;
> - simplify notation where possible and reduce unnecessary subscripts (e.g., removing the subscript $G$ in the factor $F_G$);
> - add a short intuitive overview in Section 2.2 explaining the pipeline: multi-factor block model $\rightarrow$ nodewise regression coefficients $\rightarrow$ similarity matrix $\rightarrow$ spectral clustering.
>
>
>
> **3. On how spectral clustering is implemented.**
> Thank you for raising this. We should have described the implementation more explicitly. After estimating the coefficient matrix $\mathbf{B}$, we form the symmetrized similarity matrix $\mathbf{C} = |\mathbf{B}| + |\mathbf{B}^\top|$
> and then apply the spectral clustering algorithm of Ng et al. (2001): we compute the eigenvectors of the normalized Laplacian, use them to embed the variables, and then apply $k$-means in the embedded space. This is also the standard choice in many subspace clustering methods, so using it here makes the comparison cleaner. We will state this explicitly in the revision.
>
> Reference: Ng, A., Jordan, M. and Weiss, Y., 2001. On spectral clustering: Analysis and an algorithm. Advances in neural information processing systems, 14.
>
> **4. On why $|b_{ij}|$ is used as similarity rather than, e.g., $b_{ij}^2$.**
> Our use of $|b_{ij}|$ follows standard practice in self-expressive subspace clustering methods such as SSC, SSC-EnSC, SSC-OMP, and LRR, where the magnitude of the representation coefficient is used to define the affinity matrix. We agree that replacing $|b_{ij}|$ by $b_{ij}^2$ would generally produce a different similarity matrix and hence potentially different clustering results. Our choice of $|b_{ij}|$ is therefore motivated both by standard practice in the literature and by the desire to keep our method directly comparable to the benchmark subspace clustering methods used in our experiments. We will clarify this point in the revision.
>
> **5. On whether the main contribution is better understood as learning a better similarity matrix.**
> We think this is a fair perspective. Our method, as well as standard subspace clustering methods such as SSC, SSC-EnSC, SSC-OMP, and LRR, uses a first-stage nodewise/self-expressive regression procedure to construct a similarity matrix, and then applies the same spectral clustering step to obtain the final clusters. From this perspective, differences in final clustering accuracy indeed reflect differences in the quality of the learned similarity matrices, since the downstream clustering procedure is held fixed.
>
> We chose to evaluate methods through cluster recovery accuracy, because this is the final task of interest. In addition, both our synthetic and real-data experiments come with ground-truth clusters. By contrast, there is no canonical ground-truth “similarity matrix” to compare against: the similarity matrix is only an intermediate quantity in these two-stage methods, and the same matrix may lead to different clusterings if a different second-stage algorithm is used. For this reason, we believe final clustering accuracy is the most direct and meaningful evaluation criterion here. We will clarify this rationale in the revision.

---

> > ### Author Rebuttal · Reviewer_T997 · 2026-04-05
> >
> > I am happy with the response and will raise my score.

---

### Official Review · Reviewer_A231 · 2026-03-09

**Soundness:** 4
**Presentation:** 3
**Significance:** 4
**Originality:** 4
**Overall Recommendation:** 6
**Confidence:** 4

**Summary:**

This paper addresses the problem of clustering variables in high-dimensional datasets by formulating a novel distributionally robust nodewise regression approach. The authors introduce a multi-factor “block model” for variables, which generalizes the standard single-factor (G-block) model by allowing each cluster to have multiple latent factors (so variables in the same cluster lie near a low-dimensional subspace rather than around a single centroid). Building on this model, they propose a clustering method based on a worst-case distributionally robust optimization (DRO) formulation of nodewise regression. In place of the usual $\ell_1$-regularization (as used in sparse subspace clustering), their DRO-based formulation yields a convex optimization problem with a spectral-norm regularization term. The strength of this regularizer is tied to the size of an uncertainty set in the DRO problem, which can be chosen based on the data at a given confidence level (reducing the need for manual tuning of a penalty parameter). The paper also provides a customized ADMM algorithm to solve the clustering problem efficiently, achieving significant speed-ups (several times faster) compared to off-the-shelf solvers. The proposed method is validated on both simulated data and a real-world stock returns dataset.

**Compliance With Llm Reviewing Policy:**

Affirmed.

**Key Questions For Authors:**

No

**Limitations:**

The authors could mention that their method assumes the number of clusters K is known and discuss how mis-specifying K might affect results or how one might approach choosing K.

**Strengths And Weaknesses:**

**Soundness:** The submission is technically sound and methodologically rigorous. The central claims are well-supported by both theory and experiments. The authors’ derivation of the DRO-based clustering model is correct to the best of my assessment, and they clearly state the assumptions under which the convex relaxation is valid. All theoretical results (such as conditions for the relaxation and guidelines for setting the DRO uncertainty radius) appear well-founded. The empirical evaluation is thorough and confirms the theoretical advantages, with appropriate metrics and comparisons. I did not find any significant mistakes in the proofs, algorithms, or experimental setup. In summary, the paper meets a high standard of technical soundness and correctness.

**Presentation:** The paper is generally well-written and well-structured. It provides a clear introduction and positions itself against relevant prior work (e.g., contrasting with G-block models and $\ell_1$-based clustering methods). The narrative flows logically from the problem setup, through the methodological innovation, to theoretical analysis and experiments. It includes enough detail for an expert reader to follow the derivations and (likely) reproduce the key results. The authors could mention that their method assumes the number of clusters K is known and they could discuss the possible ways of automatizing K selection. There are also a few minor typos/grammar issues that should be corrected: 1) in problem setup matrix $F$ is formally not defined, 2) the phrase "is negligible" is repeated in Appendix (line 842). These issues are small and easily fixable. Overall, the quality of writing and organization is good, and with minor revisions it can reach an excellent level.

**Significance:** This work addresses a fundamental and practical problem in machine learning: how to robustly cluster variables in high-dimensional data. The contributions have the potential for broad impact. By improving the reliability and accuracy of variable clustering (and reducing the need for manual parameter tuning), the method can benefit a variety of fields and applications. The paper’s simulation results and the finance case study both illustrate meaningful improvements over existing approaches, suggesting that researchers and practitioners could gain value from adopting this technique. Even though the proposal is somewhat specialized (focusing on clustering via nodewise regression), its implications are far-reaching within that domain – it could become a go-to method for high-dimensional clustering scenarios where robustness is required. In short, the paper advances understanding and practice in an important area, and its improvements are significant enough that they are likely to influence future research and real-world clustering tasks.

**Originality:** The paper offers a novel combination of ideas and a new perspective on clustering. To my knowledge, this is the first work to incorporate distributionally robust optimization (DRO) into the nodewise regression framework for subspace clustering. The way the authors derive a spectral-norm regularization from a worst-case (DRO) formulation is creative and goes beyond straightforward extensions of existing methods. The work differentiates itself clearly from prior literature: for example, it generalizes the single-factor block model to a multi-factor scenario and replaces heuristic $\ell_1$-penalization with a principled DRO approach. These are non-trivial innovations. Even if one views the method as a clever combination of known techniques (DRO + clustering), the outcome is a qualitatively different approach that addresses known limitations of sparse clustering in a new way. Thus, the contribution strikes me as quite original and not just an incremental improvement over prior art.

---

> ### Author Rebuttal · Authors · 2026-03-29
>
> We thank the reviewer for the very positive assessment of our work.
>
> ### **On the assumption that the number of clusters $K$ is known**
>
> We agree that this assumption should be stated more explicitly. In our pipeline, the similarity matrix is first constructed by nodewise regression, and the final partition is then obtained via spectral clustering; thus, the current method assumes that $K$ is specified at this final stage. We will make this explicit in the revision and refer to standard approaches for choosing $K$ in practice (Milligan and Cooper, 1985; Von Luxburg, 2007).
>
> We also agree that discussing misspecification of $K$ is useful. In the simulation experiments, we added an ablation for misspecification. For each $K$, we compute the adjusted mutual information (AMI) between the $K$ estimated clusters and the 25 true clusters. The performance is best at the true $K$, remains strong under moderate overestimation, and degrades more noticeably when $K$ is substantially under- or over-specified:
>
> |$K$|AMI (mean $\pm$ std)|
> |-|-|
> |10|$0.463 \pm 0.026$|
> |15|$0.682 \pm 0.032$|
> |20|$0.826 \pm 0.021$|
> |23|$0.884 \pm 0.023$|
> |**25**|**$0.920 \pm 0.025$**|
> |27|$0.919 \pm 0.020$|
> |30|$0.903 \pm 0.017$|
> |35|$0.857 \pm 0.015$|
> |40|$0.814 \pm 0.015$|
>
>
>
> Finally, note that in some applications $K$ is meant to be fixed and known. For example, in Appendix C where we study financial portfolio selection, $K$ is the number of stocks we intend to include in our (smaller) portfolio to achieve a good level of diversification compared with including all the constituents of S\&P 500.
>
>
>
> References:
>
> Milligan, G.W. and Cooper, M.C., 1985. An examination of procedures for determining the number of clusters in a data set. Psychometrika, 50(2), pp.159-179.
>
> Von Luxburg, U., 2007. A tutorial on spectral clustering. Statistics and computing, 17(4), pp.395-416.
>
> ### **On the minor presentation issues**
>
> Thank you for catching these. We will fix them in the revision.

---

> > ### Author Rebuttal · Reviewer_A231 · 2026-04-03
> >
> > Thank you, my concerns have been addressed

---

### Official Review · Reviewer_f2Ca · 2026-03-11

**Soundness:** 4
**Presentation:** 4
**Significance:** 3
**Originality:** 3
**Overall Recommendation:** 5
**Confidence:** 4

**Summary:**

This paper introduces a **Distributionally Robust Optimization (DRO)** based nodewise regression method to address the variable clustering problem under a multi-factor block model. The core innovation lies in framing traditional variable clustering as a **Subspace Clustering** problem. By constructing uncertainty sets via Wasserstein distance, the authors derive a convex optimization objective that remains robust under "worst-case" distributions. This approach effectively tackles the instability of Lasso-based nodewise regression when dealing with highly correlated variables and alleviates the difficulty of tuning the regularization parameter $\lambda$ in high-dimensional settings. Furthermore, the authors design a customized **ADMM algorithm** to bridge the gap between theoretical modeling and efficient computation.

**Compliance With Llm Reviewing Policy:**

Affirmed.

**Final Justification:**

I was thinking this is a good work and will keep my rating.

**Key Questions For Authors:**

I have detailed my specific concerns and questions within the 'Weaknesses' part of this review. I will look forward to the authors' clarifications on these points.

**Limitations:**

yes

**Strengths And Weaknesses:**

### Strengths:

- The paper establishes a rigorous mathematical chain from multi-factor models to DRO nodewise regression and subspace geometry. Theorem 3.1, in particular, offers an elegant analytical solution for addressing uncertainty in variable correlations.

- The transition from theory to practice is seamless, with the ADMM framework providing a concrete numerical solver for robust regression.

- Compared to traditional Lasso paths, the DRO approach achieves more stable "automatic parameter tuning" via a global radius $\epsilon$, showing great practical potential.

### Weaknesses:

- **Subspace Assumption**: The model relies on a linear subspace assumption. However, in real-world scenarios, non-linear correlations (e.g., quadratic or higher-order) are common. If variables lie on a non-linear manifold, linear projection might lead to clustering errors. I would appreciate it if the authors could discuss the failure boundaries of the algorithm.

- **ADMM Convergence**: While ADMM guarantees convergence for convex problems, the rate is often sensitive to the penalty parameter $\rho$. Have the authors experimented with different $\rho$ values or considered an **Adaptive $\rho$ strategy** to ensure stable convergence across diverse datasets?

- **Computational Bottleneck**: Each iteration requires **Singular Value Thresholding (SVT)**, involving a full SVD with $O(d^3)$ complexity. For high-dimensional variables (e.g., $d > 5000$), this may become a significant bottleneck. Have the authors considered using **Randomized SVD** or **Partial SVD** to accelerate the process?

---

> ### Author Rebuttal · Authors · 2026-03-29
>
> We thank the reviewer for the careful reading and positive assessment of the paper. Below we address the three concerns.
>
> ### Subspace Assumption
> > *The model relies on a linear subspace assumption. However, in real-world scenarios, non-linear correlations (e.g., quadratic or higher-order) are common. If variables lie on a non-linear manifold, linear projection might lead to clustering errors...*
>
> We agree that the current paper studies the linear multi-factor/union-of-subspaces setting. However, the framework is not inherently limited to strictly linear structure. In particular, subspace clustering can be generalized to nonlinear low-dimensional manifolds via kernel subspace clustering (Patel and Vidal, 2014), and the same idea can be applied here by replacing the nodewise regression objective with its kernelized counterpart. Following the kernel self-expressive viewpoint, one may map the variables into an implicit feature space and apply the same DRO-based regression principle there. Thus, the current method should be viewed as the linear formulation of a broader robust nodewise regression framework. When the true structure is nonlinear, a kernelized version can capture manifold geometry while preserving the robustness benefits of the DRO formulation. We will clarify this in the revision: the present paper analyzes the linear setting for theoretical clarity, while a kernel extension is a natural generalization for enhanced flexibility.
>
> Reference: Patel, V.M. and Vidal, R., 2014, October. Kernel sparse subspace clustering. In 2014 IEEE International Conference on Image Processing (ICIP) (pp. 2849-2853). IEEE.
>
> ### ADMM Convergence
> > *While ADMM guarantees convergence for convex problems, the rate is often sensitive to the penalty parameter $\rho$...*
>
> Thank you for raising this. Our ADMM implementation adopts the standard varying penalty parameter scheme in Section 3.4.1 of Boyd et al. (2010), which adaptively adjusts $\rho$ during optimization based on the primal and dual residuals. We conducted an ablation study on the sensitivity to the initial value of $\rho$. When the initial value is varied over three orders of magnitude, from
> $0.01$ to $10$,
> the clustering performance remains essentially unchanged. The following table reports summary statistics over 10 independent runs.
>
> | Initial $\rho$ | AMI (mean $\pm$ std) |
> |---|---|
> | 0.01 | $0.912 \pm 0.021$ |
> | 0.10 | $0.910 \pm 0.028$ |
> | 0.50 | $0.920 \pm 0.025$ |
> | 1.00 | $0.920 \pm 0.025$ |
> | 2.00 | $0.917 \pm 0.022$ |
> | 5.00 | $0.920 \pm 0.025$ |
> | 10.00 | $0.919 \pm 0.024$ |
>
> Therefore, the initial value of $\rho$ is not a delicate tuning knob in our implementation because we use an adaptive update and observe very small performance variation across a wide initialization range. We will add the sensitivity analysis to the revised manuscript.
>
> Reference: Boyd, S., Parikh, N., Chu, E., Peleato, B. and Eckstein, J., 2010. Distributed optimization and statistical learning via the alternating direction method of multipliers. Foundations and Trends in Machine Learning, 3(1), pp.1-122.
>
> ### Computational Bottleneck
> > *Each iteration requires Singular Value Thresholding (SVT), involving a full SVD with $O(d^3)$ complexity...*
>
> We agree this is an important practical point. The current ADMM update for the spectral-norm term indeed relies on singular value decomposition in the $B_2$ step, via the partially closed-form solution in Lemma 3.2. The paper is transparent about this structure: the $B_2$ update is obtained by computing the singular value decomposition of $I - B_1^{t+1} - \Lambda^t$, and this is the key spectral operation inside the algorithm.
> The present experiments demonstrate scalability to settings such as $d=500$ in simulation and $d=504$ in the face-clustering experiment, where the method performs strongly and the ADMM solver is practical. The ADMM algorithm reduces the runtime of the DRO clustering method by over 80\% compared to off-the-shelf convex optimizers and is competitive with other clustering algorithms. See Table 8 in Appendix D.7 of the initial submission.
>
> For much larger dimensions, e.g. $d > 5000$, one should consider approximate spectral routines such as randomized SVD (Halko et al., 2011; Tropp et al., 2017). Because the $B_2$ step depends on the singular spectrum of a structured matrix and effectively shrinks only the top singular values, such approximations are a natural engineering improvement and are compatible with the algorithmic framework. We will add this to the revised manuscript as a concrete direction for scaling the method further.
>
> References:
>
> Halko et al., 2011. Finding structure with randomness: Probabilistic algorithms for constructing approximate matrix decompositions. SIAM review.
>
> Tropp et al., 2017. Practical sketching algorithms for low-rank matrix approximation. SIAM Journal on Matrix Analysis and Applications.

---

> > ### Author Rebuttal · Reviewer_f2Ca · 2026-04-03
> >
> > I have no more questions for this work

---

### Decision · Program_Chairs · 2026-04-30

**Decision:**

Accept (regular)

**Comment:**

This manuscript considers the variable clustering problem and its connections to subspace clustering. A convex relaxation is provided along with substantial numerical results. The reviews collectively identified that the manuscript has several key strengths: it addresses and important problem, blends a set of distinct ideas into a nice framework, and has strong theoretical and numerical results. While a few, minor shortcomings were identified, the strengths outweigh those weakness and lead to the recommendation.